# Multipotent versus differentiated cell fate selection in the developing *Drosophila* airways

Ryo Matsuda[1], Chie Hosono[1], Christos Samakovlis[1,2,3]*, Kaoru Saigo[4]

[1]Department of Molecular Biosciences, The Wenner-Gren Institute, Stockholm University, Stockholm, Sweden; [2]Science for Life Laboratory, Solna, Sweden; [3]ECCPS, Justus Liebig University of Giessen, Giessen, Germany; [4]Department of Biophysics and Biochemistry, Graduate School of Science, University of Tokyo, Tokyo, Japan

**Abstract** Developmental potentials of cells are tightly controlled at multiple levels. The embryonic *Drosophila* airway tree is roughly subdivided into two types of cells with distinct developmental potentials: a proximally located group of multipotent adult precursor cells (P-fate) and a distally located population of more differentiated cells (D-fate). We show that the GATA-family transcription factor (TF) Grain promotes the P-fate and the POU-homeobox TF Ventral veinless (Vvl/Drifter/U-turned) stimulates the D-fate. Hedgehog and receptor tyrosine kinase (RTK) signaling cooperate with Vvl to drive the D-fate at the expense of the P-fate while negative regulators of either of these signaling pathways ensure P-fate specification. Local concentrations of Decapentaplegic/BMP, Wingless/Wnt, and Hedgehog signals differentially regulate the expression of D-factors and P-factors to transform an equipotent primordial field into a concentric pattern of radially different morphogenetic potentials, which gradually gives rise to the distal-proximal organization of distinct cell types in the mature airway.

*For correspondence: Christos. Samakovlis@su.se

**Competing interests:** The authors declare that no competing interests exist.

## Introduction

Multipotent stem cells are essential for both growth and generation of cell diversities in developing organs. They also serve as reservoir to replace damaged or aged cells during physiological or pathological tissue homeostasis. The discovery of a small number of transcription factors that can induce pluripotent cells (*Takahashi and Yamanaka, 2006*) has fueled major research efforts to reveal the mechanisms biasing the choice between pluripotent/multipotent and more differentiated cell fates in vivo (*O'Brien and Bilder, 2013*). Using the *Drosophila* airways (tracheal system) (*Manning and Krasnow, 1993*; *Samakovlis et al., 1996b*) (*Figure 1A*), we studied how the initial selection of different potencies within an organ becomes first predisposed and then regionally confined.

By mid-embryogenesis, *Drosophila* embryos are metamerically divided along the anterior-posterior (AP) axis (*Ingham, 1988*; *Sanson, 2001*). Each unit (segment or parasegment) is also subdivided along the dorso-ventral (DV) embryo axis (*Anderson et al., 1985b*; *Wharton et al., 1993*). Generation of the airway primordia is spatially restricted along these AP or DV axis, by the local concentrations of repressing and activating signals (*Bier et al., 1989*; *Isaac and Andrew, 1996*; *Perrimon et al., 1991*; *Wilk et al., 1996*). For example, Wingless (Wg) expressed in stripes along the AP axis and Decapentaplegic (Dpp) expressed in the dorsal ectoderm (*Figure 1B*) repress the airway primordia specification (*Isaac and Andrew, 1996*; *Wilk et al., 1996*). As a result, 10 metameric groups of cells expressing a master gene, Trachealess (Trh) are specified on each side of the lateral ectoderm (*Figure 1A–B*). Invagination of each primordial cluster transforms the two-

**eLife digest** Many organs are composed of tubes of different sizes, shapes and patterns that transport vital substances from one site to another. In the fruit fly species *Drosophila melanogaster*, oxygen is transported by a tubular network, which divides into finer tubes that allow the oxygen to reach every part of the body.

Different parts of the fruit fly's airways develop from different groups of tracheal precursor cells. P-fate cells form the most 'proximal' tubes (which are found next to the outer layer of the fly). These cells are 'multipotent' stem cells, and have the ability to specialize into many different types of cells during metamorphosis. The more 'distal' branches that emerge from the proximal tubes develop from D-fate cells. These are cells that generally acquire a narrower range of cell identities.

By performing a genetic analysis of fruit fly embryos, Matsuda et al. have now identified several proteins and signaling molecules that control whether tracheal precursor cells become D-fate or P-fate cells. For example, several signaling pathways work with a protein called Ventral veinless to cause D-fate cells to develop instead of P-fate cells. However, molecules that prevent signaling occurring via these pathways help P-fate cells to form. Different amounts of the molecules that either promote or hinder these signaling processes are present in different parts of the fly embryo; this helps the airways of the fly to develop in the correct pattern.

This work provides a comprehensive view of how cell types with different developmental potentials are positioned in a complex tubular network. This sets a basis for future studies addressing how the respiratory organs – and indeed the entire organism – are sustained.

dimensional (2D) shape of each cluster to 3D tubes (*Figure 1C*) (*Campos-Ortega and Hartenstein, 1985*; *Turner and Mahowald, 1977*). Within the invaginated primitive sacs/tubes, the proximal cells form a narrow cord, the spiracular branch/SB (P-fate) (*Figure 1C, K*) and stay multipotent to later proliferate and to generate many parts of the pupal/adult airways (*Manning and Krasnow, 1993*; *Pitsouli and Perrimon, 2010*; *Weaver and Krasnow, 2008*). Within the remaining distal cells (D-fate), the more distal parts achieve a series of morphogenetic events, extending six primary branches, fusing with branches from neighboring metameres and supplying oxygen directly to the target cells (*Manning and Krasnow, 1993*; *Samakovlis et al., 1996a*; *Samakovlis et al., 1996b*), while the more proximal parts (transverse connectives/TC) connect SB with the primary branches (*Figure 1C, K*).

The primary branches constitute D-fate cells located distally from TC, and their morphological diversification depends on the activation of various signaling pathways. For example, *rhomboid (rho)* is expressed in each primordium (*Bier et al., 1990*) and activates *dEGFR* (also called *torpedo* or *faint little ball*) (*Price et al., 1989*; *Schejter and Shilo, 1989*) by generating the secreted active ligand Spitz (s-Spi) (*Gabay et al., 1997*; *Golembo et al., 1996*; *Schweitzer et al., 1995b*; *Urban et al., 2001*; *Wappner et al., 1997*). Activated dEGFR instructs the cytoskeletal changes that coordinate invagination (*Brodu and Casanova, 2006*; *Kondo and Hayashi, 2013*; *Llimargas and Casanova, 1999*; *Nishimura et al., 2007*). Branchless (Bnl)/dFGF is expressed in patches of surrounding tissues to guide primary branching (*Sutherland et al., 1996*) by activating Breathless (Btl)/dFGFR on the airway cells (*Klambt et al., 1992*; *Lee et al., 1996*). Apart from receptor tyrosine kinase (RTK) signaling, Decapentaplegic (Dpp)/BMP, Wingless (Wg)/WNT, and Hedgehog (Hh) signals combinatorially establish different primary branch identities (*Affolter et al., 1994*; *Chen et al., 1998*; *Chihara and Hayashi, 2000*; *Glazer and Shilo, 2001*; *Llimargas, 2000*; *Llimargas and Lawrence, 2001*; *Matsuda et al., 2015*; *Vincent et al., 1997*). Although many studies focused on the differentiation and subsequent morphogenesis of the D-fate cells (*Beitel and Krasnow, 2000*; *Cabernard et al., 2004*; *Ghabrial et al., 2011*; *Hosono et al., 2015*), generation of the P-fate has been underexplored. Our current dissection of P/D fate establishment in the embryo, together with metamere-specific control programs of differentiation/de-differentiation in the larval airways (*Djabrayan et al., 2014*; *Guha et al., 2008*; *Sato et al., 2008*; *Weaver and Krasnow, 2008*) may broaden our understanding of the strategies for stem cell selection and maintenance in vivo.

## Results and discussion

### The proximo-distal cell fate organization in the *Drosophila* airway tree

Every cell in the main airways derives from 20 primordial cell clusters expressing the master regulator TF Trh (*Figure 1A–C, K*) (*Isaac and Andrew, 1996*; *Perrimon et al., 1991*; *Wilk et al., 1996*). However, the expression of several other genes is enriched either distally or proximally in the airways. *rho* is expressed preferentially in the central and distal regions of the airway primordia before and during invagination (*Figure 1—figure supplement 1A–E*). *btl/dFGFR* is also expressed preferentially in the distal part (*Figure 1D–E*), and its expression is upregulated by Bnl/dFGF in the most distal leading cells of the primary branch tips (*Figure 1E*) (*Ohshiro et al., 2002*; *Ohshiro and Saigo, 1997*). mAb2A12 detecting the putative chitin-biding protein Gasp (*Tiklova et al., 2013*) preferentially labels the distal airways during stages 14-17 (*Figure 1F*) (*Samakovlis et al., 1996a*). On the other hand, *unpaired (upd)*, a ligand activating the JAK/STAT signaling is expressed only in the proximal SB cells from stage 12 on (*Figure 1G–I*) (*Harrison et al., 1998*). From stage 13 onwards, the SB cells also show stronger expression of the *P0144* enhancer trap marker (*Figure 1E,F,I*) (http://fly-view.uni-muenster.de). Compared to the anterior metameres, the 10th tracheal primordium (Tr10) does not establish the *P0144*-postive P-fate due to the high level of a Hox protein Abdominal-B (AbdB) in this part of the embryo (*Figure 1F, J*), (*Celniker et al., 1989*; *Matsuda et al., 2015*). In *trh* mutants, both the D-fate marker (*Chung et al., 2011*; *Jin et al., 2001*; *Ohshiro and Saigo, 1997*; *Zelzer and Shilo, 2000*) and the P-fate marker expressions (*Figure 1—figure supplement 1U, V*) are lost.

To define the origins of the proximo-distal organization of the airway tree, we labeled cell groups during the 2D primordial stage and recorded their fates in the 3D tree. *pointed (pnt)* is a general transcriptional target and mediator of RTK signaling (*Brunner et al., 1994*; *Klambt, 1993*; *O'Neill et al., 1994*). *rho* induces *pnt* expression in the central parts of the airway primordia (*Figure 1—figure supplement 1F–O*) and *btl/dFGFR* upregulates *pnt* expression in the tips of the primary branches (*Figure 1—figure supplement 1M–O*) (*Samakovlis et al., 1996a*). Markers of the central parts of the airway primordia (*pnt-lacZ* and *btl-CD8-GFP*) become preferentially expressed in the distal part of the airway tree (*Figure 1K*, *Figure 1—figure supplement 1N–O*). Similarly, *hairy (h)* expression in the airway primordia (*Carroll et al., 1988*; *Hooper et al., 1989*; *Zhan et al., 2010*) is confined to the dorso-central part (*Figure 1—figure supplement 1P*) and its reporter (*h-lacZ*) becomes active in most of the DB, VB, and DT cells but not in TC, SB nor in ventral branches (*Figure 1K*, *Figure 1—figure supplement 1Q*) (*Samakovlis et al., 1996a*). In contrast, *salm* labels the dorsal part of the airway primordia (*Figure 1—figure supplement 1R*), (*Kuhnlein and Schuh, 1996*) and *salm-gal4* driven *UAS-GFP* marks DB and DT, and parts of TC and SB in the airways (*Figure 1K*, *Figure 1—figure supplement 1S–T*). These results suggest that the dorso-peripheral sector of each primordium, which is positive for *salm* but lacking *h*, composes half of the proximal SB cells in the airway tree. The other half may originate from the ventro-peripheral sector of the primordia. Consistently, around the completion of invagination, the P-fate marker *upd* is enriched in a horseshoe-like peripheral ring (*Figure 1G*)(*Harrison et al., 1998*), which eventually marks the SB cells at stage 12 (*Figure 1H,I,K*). Collectively, the gene expression analysis suggests a fate map of the airway primordia, where the centro-peripheral axis inversely correlates both with the ordered pattern of tracheal cell invagination (*Brodu and Casanova, 2006*; *Nishimura et al., 2007*) and the PD axis of airway branching (*Figure 1C,K*).

### RTK signaling drives the D-fate selection at the expense of the P-fate

As activation of RTKs propels several morphogenetic programs in the D-cells, we first assessed their effects on the discrimination of the P- and the D-fate. In *btl/dFGFR* mutants, where active branch extension does not occur, apoptotic staining in the distal cells increases from stage 12 (*Figure 2—figure supplement 1A,B*), suggesting that *btl/dFGFR* signaling directly or indirectly suppresses apoptosis of the distal airways. In *dEGFR btl/dFGFR* double mutants, massive cell death staining accompanies loss of most tracheal cells compared to either single mutant (*Figure 2—figure supplement 1B–D*). Suppression of apoptosis by removing pro-apoptotic genes using *Df(H99)* (*Figure 2—figure supplement 1E*) (*White et al., 1994*) or by transgenic expression of the apoptosis inhibitor, DIAP (*Figure 2—figure supplement 1F*) (*Hay et al., 1995*) partially restores the survival of tracheal cells

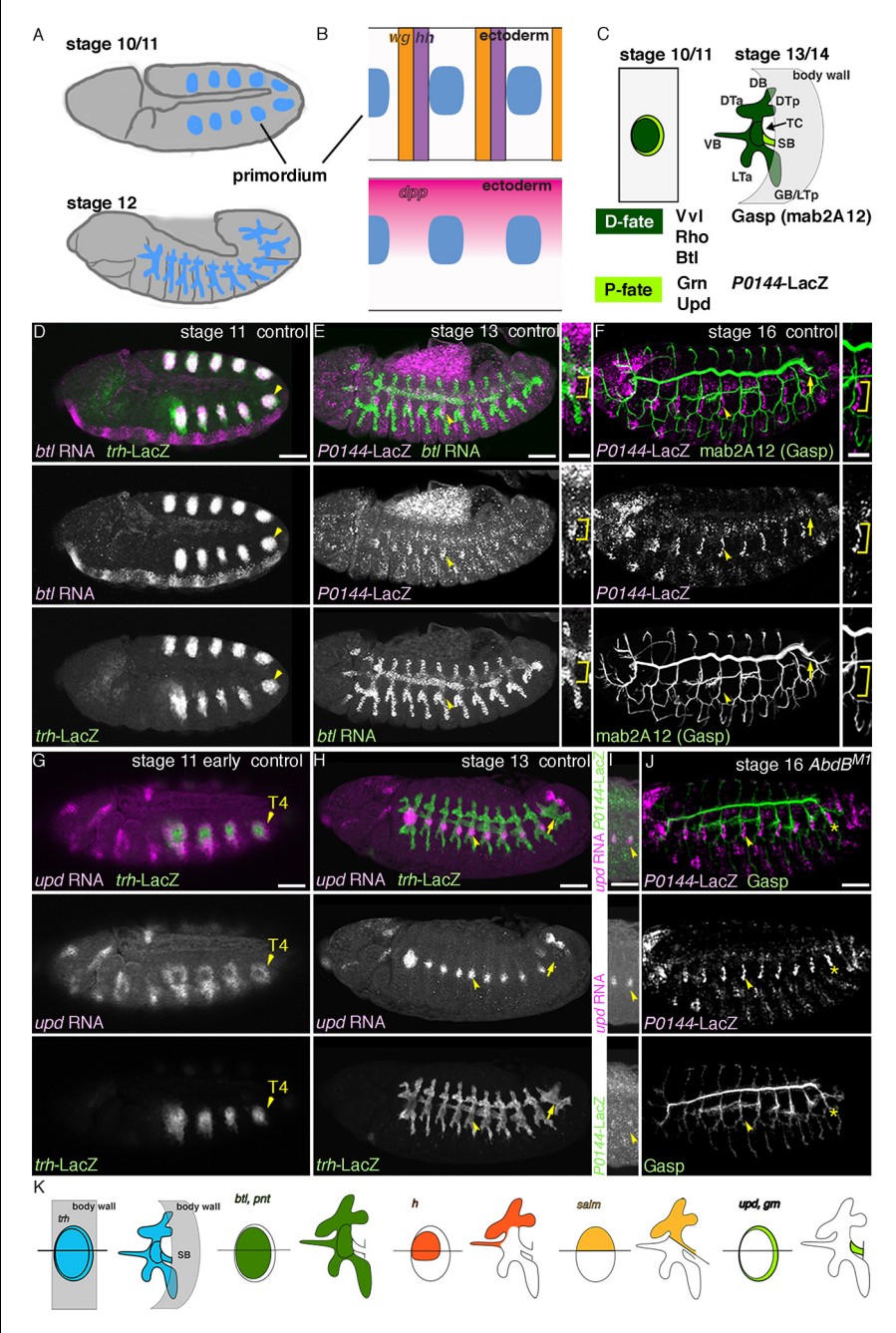

**Figure 1.** The proximo-distal cell fate organization of the *Drosophila* airways. (**A**) A sketch of the embryonic *Drosophila* airways at stage 10/11 and late stage 12. (**B**) A representation of the ectodermal expression the secreted signaling molecules *wg/WNT, hh* (upper panel), and *dpp* (lower panel) in relation to the airway primordia at stage 10. (**C–J**) Expression of the P/D-fate markers at different stages of the airway development, which is summarized schematically in **C**, where the regional diversification of the *Drosophila* airway according to the proximo-distal axis is also shown in different colors. The P-fate region: spiracular branch (SB). The D-fate region: transverse connectives (TC) and six primary branches (dorsal branch/DB, dorsal trunk anterior/DTa, dorsal trunk posterior/DTp, visceral branch/VB, lateral trunk anterior/LTa, and ganglionic branch/GB/lateral trunk posterior/LTp). Expression of the D-fate markers, *btl* (D,E), mab2A12 (**F, J**) and the P-fate markers, *P0144-lacZ* (**E, F, I, J**) and *upd* (**G-I**), relative to each other or *trh-lacZ* (**D, G, H**). In this and other figures, arrowheads mark one of the 10 metameres. In **E-F**, enlarged picture of Tr5 is also shown, where the P-fate cells are bracketed. *btl* expression at stage 11 (**D**) is concentrated in the central parts of the primordia. At stage 13 (**E**), *btl* expression and *P0144-lacZ* expression (arrowheads) rarely overlap. At stage 16, mab2A12 (anti-Gasp antibody) strongly labels the lumens of

*Figure 1 continued on next page*

*Figure 1 continued*

the D-cells while *P0144-lacZ* is strongly detected in the P-cells (**F**, arrowheads). At stage 11 (**G**), *upd* transcript is detected at the peripheral area of each primordium (arrowheads). At stage 13 (**H, I**), *upd* is expressed in the P-cells that express *P0144-lacZ* (arrowheads). Note that compared to the control, where *upd* and *P0144-lacZ* are not expressed in Tr10 (arrows in **F, H**), the P-fate (*P0144-lacZ*) is established in Tr10 of *AbdB* mutants (asterisks in **J**). Scale bar is 2 μm in the enlarged panels of **E-F**. Scale bar in the remaining panels is 50 μm. (**K**) summarizes typical gene expression patterns of various marker genes in the airway primordium and in the mature airways. See text for details.

The following figure supplement is available for figure 1:

**Figure supplement 1.** The centro-peripheral organization of the airway primordia.

in *dEGFR btl/dFGFR* double mutants (*Figure 2—figure supplement 1E–J*). Thus, dEGFR and Btl/dFGFR cooperate to suppress apoptosis in the airway cells. To analyze the effects of RTK signaling in the P/D fate choice independently of cell survival, we introduced *Df(H99)* into some of the genotypes used in the analysis described below.

In *bnl/dFGF* mutants, the feedback upregulation of the D-fate marker *btl/dFGFR* is lost in the tips of the primary branches (*Figure 2A,B*) (*Ohshiro et al., 2002*). However, the distal part of the invaginated stump retains the mab2A12 signal (*Lee et al., 1996*) and the basal *btl/dFGFR* expression (*Figure 2B*). Similarly, expression of the P-fate markers (*upd* and *P0144-lacZ*) in *btl/dFGFR* mutants remains largely comparable to the wild type (*Figure 2E–F, I–J, M*). These results suggest that Btl/dFGFR signaling does not play a major role in the P-D discrimination. In *rho* mutants, however, the refinement of *upd* expression to the primordia periphery is retarded (*Figure 2—figure supplement 1K,L*), and the expression domains of *upd* and *P0144-lacZ* become expanded (*Figure 2G, K, M*). Correspondingly, the domain of *btl*-expressing cells is decreased (*Figure 2C*) suggesting that the *rho*-mediated *dEGFR* signaling drives the D-fate selection at the expense of the P-fate. We investigated the possible redundant functions of the two RTKs in D-fate definition. In *rho bnl/dFGF* double mutants, *btl* expression is further reduced or even lost in some segments (*Figure 2D*). Correspondingly, in *rho btl/dFGFR* double mutants, another D-fate marker mab2A12/Gasp becomes reduced or undetectable in some segments, accompanied by a concomitant increase of the proximal *P0144-lacZ* expression domain (*Figure 2L, N*). Consistent with the phenotypes of *rho btl/dFGFR* mutants, *P0144-lacZ* expression is also increased in *pnt* mutants, (*Figure 2M*, *Figure 2—figure supplement 1P*). Taken together, we conclude that *dEGFR* signaling in cooperation with *btl/dFGFR* signaling promotes the D-fate selection at the expense of the P-fate partly through *pnt* (*Figure 2O*). The residual weak expression of the D-fate markers in *rho btl* or *dEGFR btl/dFGFR* double mutants (*Figure 2L*, *Figure 2—figure supplement 1M–O*) suggests additional inputs in the D fate selection, independent of RTK signaling downstream of *dEGFR* and *btl/dFGFR* (*Figure 2O*).

## vvl drives the D-fate selection at the expense of the P-fate

The expression of two key components of RTK signaling, *rho* in the central part of the airway primordia and *btl* in the extending distal airways, partly depends on the POU-domain transcription factor *vvl* (*Anderson et al., 1996*; *Llimargas and Casanova, 1997*). We confirmed that in *vvl* mutants, expression of the D-fate markers *rho* and *btl* is reduced (*Figure 3A, B, E, F*). Moreover, mAb2A12 staining is almost diminished (*Figure 3N, O*) in the residual distal branches, which is restored by *UAS-vvl* driven with *btl-gal4* (*Figure 3—figure supplement 1T*). By contrast, the expression of the P-fate marker *upd* is expanded to the distal branches at stage 12 (*Figure 3J–K*) and the *P0144-lacZ* domain is significantly increased at stage 15 in *vvl* mutants (*Figure 2N*, *3O*). This indicates that *vvl* promotes the D-fate at the expense of the P-fate, and a part of this function may be achieved through *vvl*-dependent expression of *rho* and *btl*. In wild type, *vvl* expression is enriched in the distal trachea, indicating that *vvl* itself is a D-fate marker (*Figure 3—figure supplement 1A–C*).

The residual expression of the RTK signaling components in *vvl* mutant may function as D-factors in the absence of *vvl*. To evaluate the relative contributions of *vvl* and RTKs in the P/D fate selection, we examined double or triple mutants of *vvl* and components of the two RTK signaling pathways. Compared to single or double mutants (*Figure 3—figure supplement 1E, F*), in *vvl rho bnl* triple

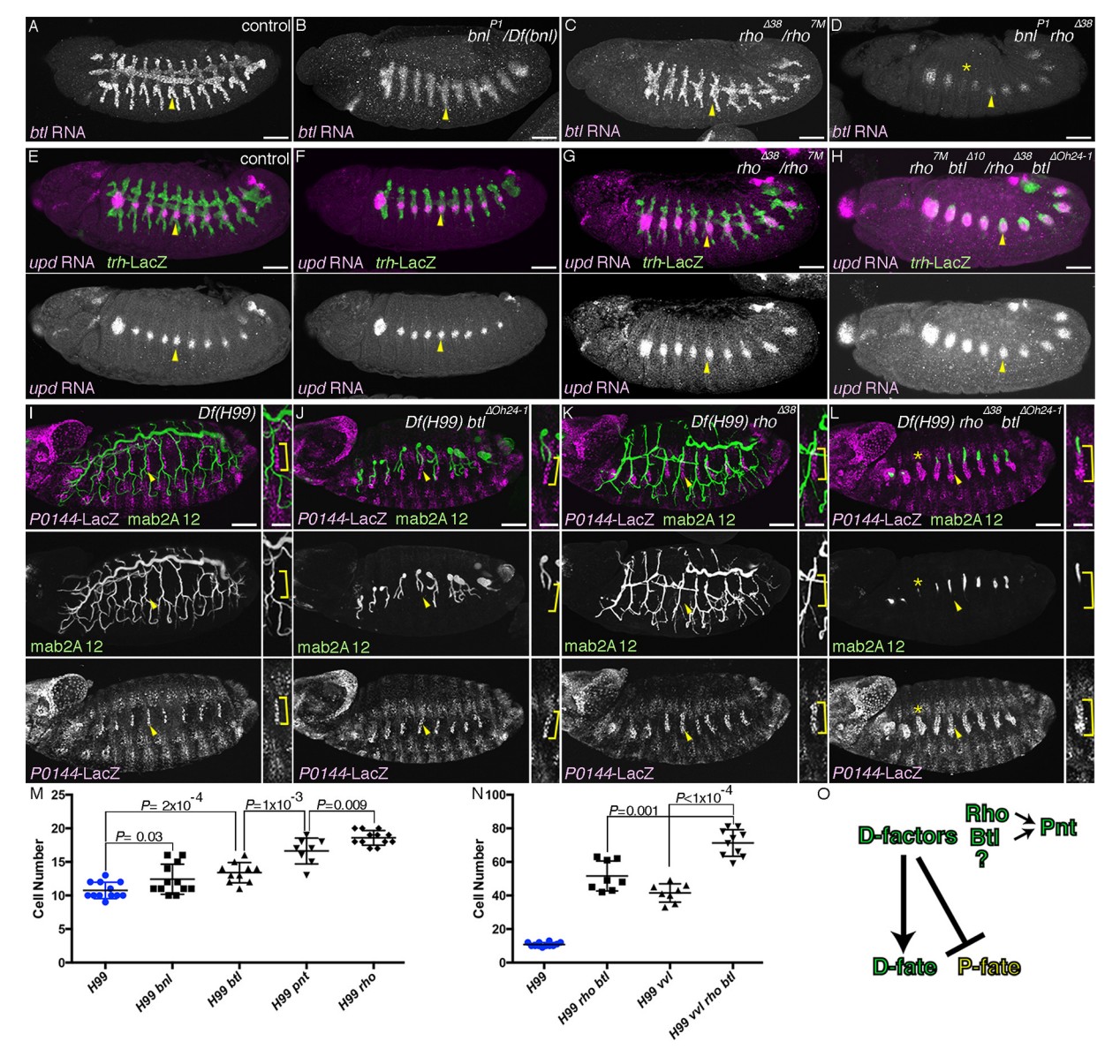

**Figure 2.** RTK activation drives the D-fate selection. Expression of the D-fate (*btl* in A-D and mab2A12 in I-L) and the P-fate (*upd* in E-H and *P0144-lacZ* in I-L) markers upon loss of *dEGFR* and/or *btl/dFGFR* signaling. (A–H) Stage 12/13. (I–L) Stage 16. Compared to the control (A, E, I), in *bnl* mutants (B), *btl* mutants (F) or *Df(H99) btl* mutants (J), expression of the D-fate markers *btl* (A) and mab2A12 (J) and the P-fate markers *upd* (F) and *P0144-lacZ* (J) appears similar. Note, however, that upregulation of *btl* in a pan-tip pattern is abolished in *bnl* mutants (B) (**Ohshiro et al., 2002**). In *rho* mutants (C, G) or *Df(H99) rho* mutants (K), the expression area of the P-fate markers *upd* (G) and *P0144-lacZ* (K, arrowheads) is increased, leading to the decreased area of the D-fate marker expression (C, K). In *rho bnl* double mutants (D), *rho btl* double mutants (H) or *Df(H99) rho btl* triple mutants (L), the P-fate area is further expanded with concomitant reduction of the D-fate markers. Note that in D,L expression of the D-fate markers becomes reduced very much in a few segments (asterisk). Scale bar is 2 μm in the enlarged panels of I-L where the P-fate cells are bracketed. Scale bar in the remaining panels is 50 μm. (M,N) Scatter plots of the number of *P0144-lacZ* positive P-fate cells in Tr5 and Tr6 of the indicated genotypes at stage 16. All p-values were calculated by Student's t-test. Source files are supplied in *Figure 2—source data 1,2*. (O) A scheme of P/D-fate selection by RTK signaling components. RTK, receptor tyrosine kinase.

The following source data and figure supplement are available for figure 2:

**Source data 1.** Source data for *Figure 2M*.
**Source data 2.** Source data for *Figure 2N*.

*Figure 2 continued on next page*

*Figure 2 continued*

**Figure supplement 1.** Multiple roles of *dEGFR* and *btl/dFGFR* signaling.

mutant embryos, the expression of the D-fate marker *btl* is initiated at stage11 but is not maintained by stage 12 (*Figure 3—figure supplement 1G,H*). Consistently, the mab2A12 signal is undetectable in *vvl rho btl* mutants (*Figure 3J–L*). In contrast, expression of the P-fate markers *upd* and *P0144-lacZ* appears to be expressed in all tracheal cells (*Figure 2N*, *Figure 3—figure supplement 1I, L*). These results suggest that the two RTK pathways act both downstream of and in parallel to *vvl* to drive the D-fate selection at the expense of the P-fate.

## Hh and Vvl drive the D-fate selection

Hh is expressed just anteriorly to the airway primordia (*Glazer and Shilo, 2001*) and is proposed to enhance the primordial *vvl* expression (*Llimargas and Casanova, 1997*). We confirmed that in *hh* mutants, *vvl* expression is variably decreased or lost (*Figure 3—figure supplement 1D*). Consistently, expression of the D-fate markers *rho* and *btl* is also variably decreased or lost in *hh* mutants (*Figure 3C,G*, in 6 out of 26 embryos for *btl* expression). In contrast, expression of the P-fate marker *upd* is expanded and sometimes occupies the whole trachea in some segments of *hh* mutants (*Figure 3L*). A similar effect is observed on the expression of a different set of the P/D fate markers, mab2A12/Gasp and *P0144-lacZ* (*Figure 3P*). These results suggest that Hh signals from the anterior border of the airway primordia to orient the P/D fate selection toward the D-fate direction. Similar to the cooperative activities of *rho, btl,* and *vvl, hh* could synergize with *vvl* to drive the D-fate. Indeed, in the absence of both *hh* and *vvl*, although expression of the D-fate markers *rho* and *btl* is variably detected in the primordia (*Figure 3D,H*), *btl* expression is completely absent later (*Figure 3I*). Correspondingly, the expression of P-fate markers expands to all tracheal cells (*Figure 3M,Q*). A similar trend was observed in mutants for *hh* and RTK signaling components (*Figure 3—figure supplement 1M–S*). Taken together, we conclude that Hh and Vvl cooperatively drive the D fate selection, partly through the activation of *dEGFR* and *btl/dFGFR* signaling in the central part of the primordium and in the distal branches.

## Grn promotes the P-fate selection

The identification of several signaling pathways converging in the initiation and establishment of the D-fate in the airways prompted us to interrogate what promotes the P-fate. We revealed the GATA-family TF Grn (*Brown and Castelli-Gair Hombria, 2000*; *Garces and Thor, 2006*; *Lin et al., 1995*) as the P-fate promoting factor. *grn* is preferentially expressed in the peripheral parts of the airway primordia from stage 11 on (*Figure 4—figure supplement 1A*). At stage 13, the P-fate cells of the SB and the overlying lateral ectoderm are positive for *grn* (*Figure 4—figure supplement 1B–C*). *grn* expression is repressed in the D-cells by the D-fate determinants, *hh, vvl,* and RTKs (*Figure 4—figure supplement 1D–G*), while *grn* expression in the lateral ectoderm is largely intact in the absence of tracheal cells in *upd* or *trh* mutants (*Figure 4—figure supplement 1H–I*). Grn overexpression results in the upregulation of the P-fate marker *P0144-lacZ* in the more distal TC branches (*Figure 4A–C*). Conversely, in *grn* mutants, expression of the P-fate markers *upd* and *P0144-lacZ* is lost (*Figure 4D–F*, *Figure 4—figure supplement 1J*), resulting in mAb2A12-positive tubes directly connecting to the epidermis. Counting the Trh-positive cells in Tr5 and in TC5/SB5 reveals that around 10 cells are selectively lost from the TC/VB region (*Figure 4G*). As halving of the airway cell number in CycA mutants (*Beitel and Krasnow, 2000*) does not abolish the P-fate (*Figure 4H*, *Figure 4—figure supplement 1K*), we suggest that reduction of cell number does not account for the loss of the P-fate in *grn* mutants. Also, *arm-gal4*-mediated overexpression of Trh in *grn* mutants does not restore the P-fate (*Figure 4H*, *Figure 4—figure supplement 1L*). These results suggest that *grn* functions in two ways, namely to establish Trh expression in the P-region and to induce the P-fate marker expression (*Figure 4K*). Accordingly, we designate *grn* a master regulator of the P-fate.

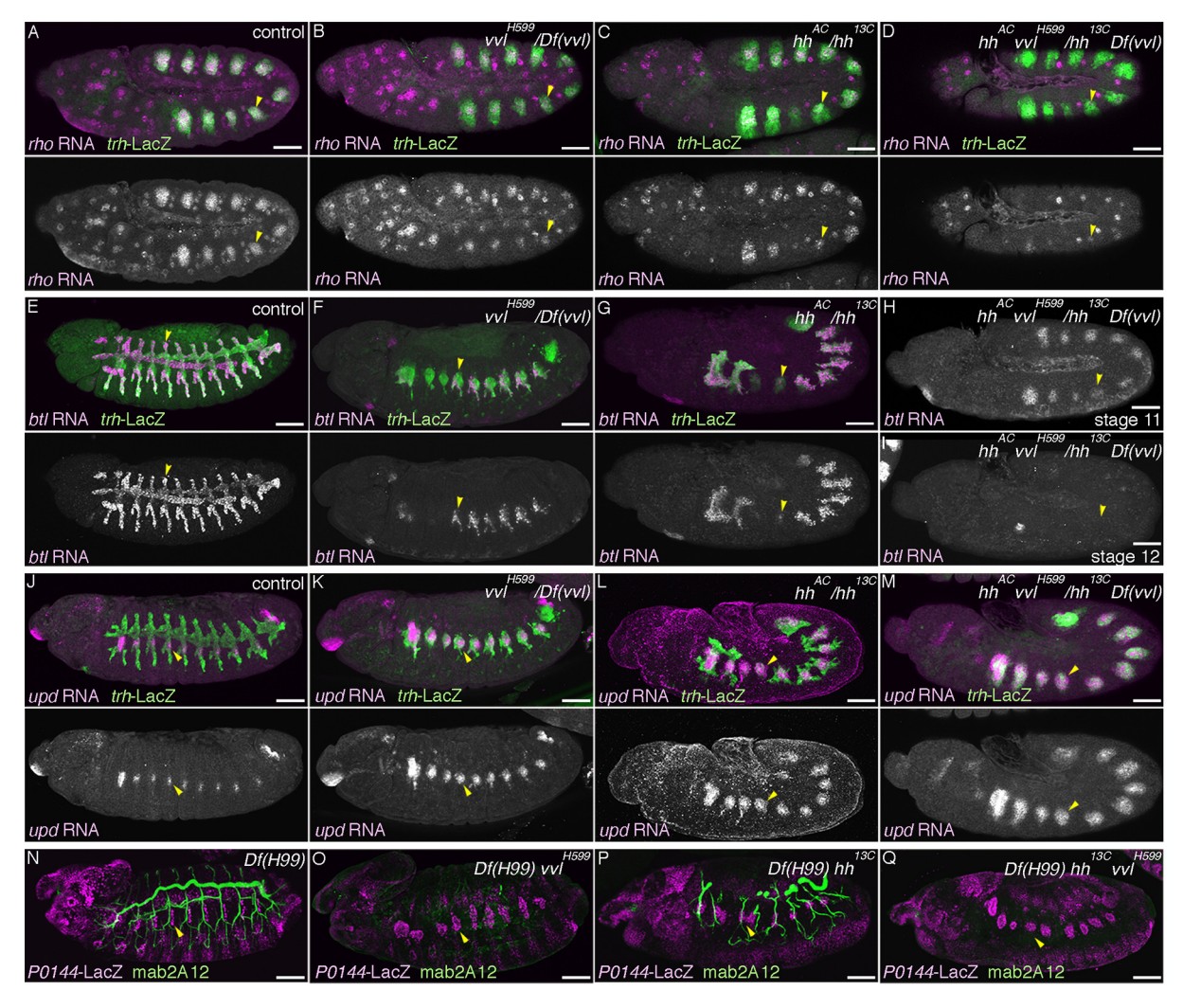

**Figure 3.** Hh and Vvl drive the D-fate selection. Expression of the D-fate markers (*rho* at stage 11 in **A–D**, *btl* at stages 11-13 in **E–I** and mab2A12 at stages 15/16 in **N–Q**) and the P-fate markers (*upd* at stages 12/13 in **J–M**, *P0144-lacZ* in **N-Q**) upon loss of *vvl* and/or *hh*. Note that the genotypes in (**N-Q**) additionally carry the *Df(H99)* mutation. Compared to the control (column 1), upon loss of *vvl* (column 2), the expression area of the D-fate markers *rho* (**B**), *btl* (**F**), and mab2A12 (**Q**) is decreased while expression of the P-fate markers *upd* (**K**) and *P0144-lacZ* (**Q**) is expanded. Note that mab2A12 signals are hardly detectable in **O**. *hh* mutants (column 3) show the same trend as *vvl* mutants. Expression of *btl* (**G**) and mab2A12 (**P**) is variably lost in Tr3/4 and *upd* expression (**P**) expands to the whole distal area. In *hh vvl double* mutants (column 4), expression of the D-fate markers (**D,I**) is lost, while expression of the P-fate markers (**M,Q**) persist in the whole trachea. Note that weak *btl* expression is detectable in the primordia (**H**) but is lost soon after (**I**). Scale bar: 50 μm.

The following figure supplement is available for figure 3:

**Figure supplement 1.** Hh, Vvl, and RTKs collaborate to drive the D-fate selection.

## Effects of simultaneous loss of the D- and P-factors

Having identified several regulators of the D- and P-fates, we examined their epistatic relationships. Compared to the control (*Figure 5A*, *Figure 5—figure supplement 1E, J*), in *grn hh* double- (*Figure 5C*, *Figure 5—figure supplement 1G, L*) or *grn rho btl* triple mutants (*Figure 5—figure supplement 1I, N*), the P-fate marker expression is not robustly restored. In *grn vvl* double mutants, the *trh* expression area appears to be very much reduced (*Figure 5F*), probably because the P-fate cells depend on *grn* for *trh* expression. However, in this reduced area of the distal region, we

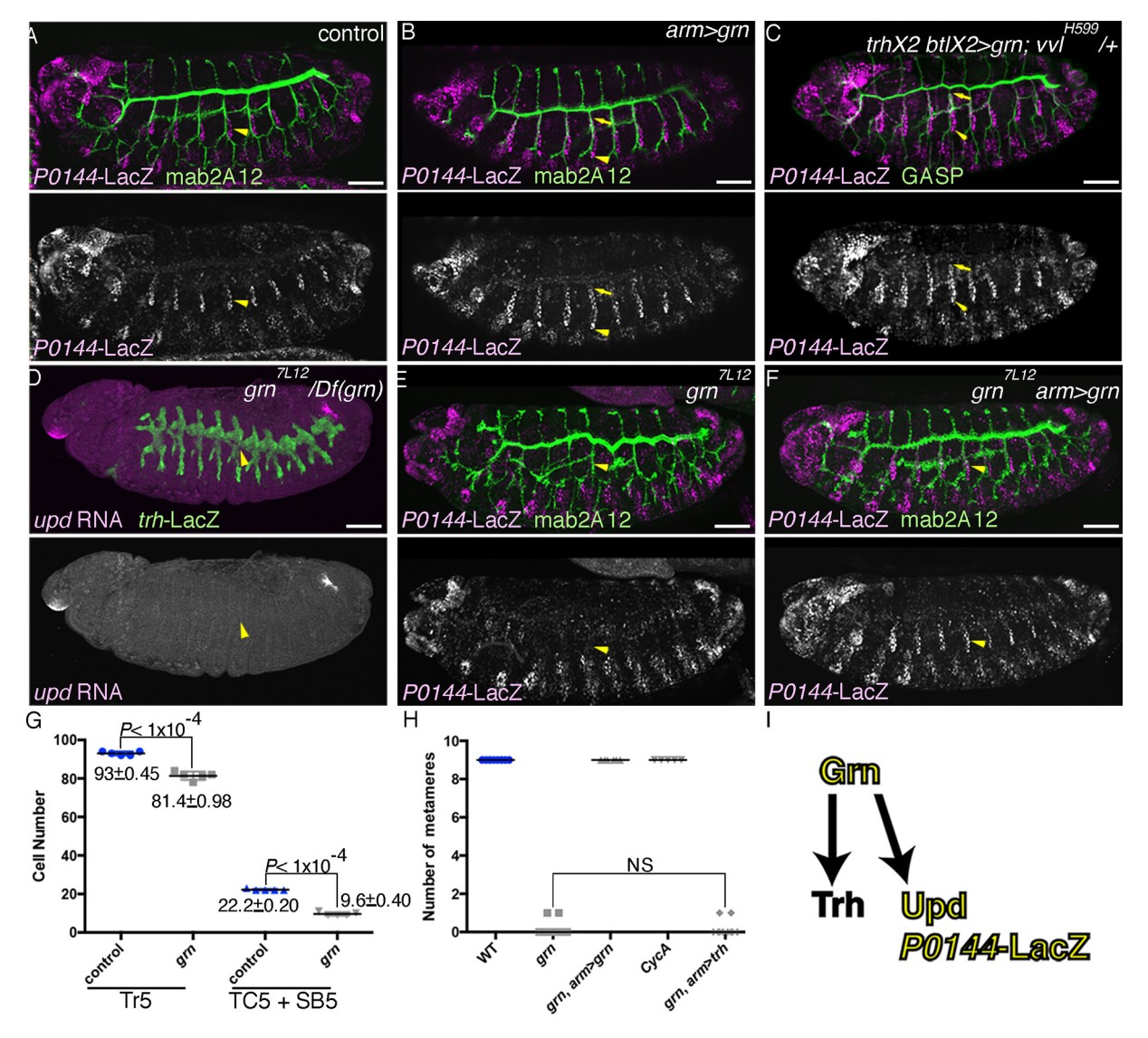

**Figure 4.** Grn promotes the P fate. Compared to the control (**A**), *grn* overexpression with *arm-gal4* (**B**) or two copies of *btl-gal4* and *trh-gal4* (**C**) slightly expands the area of *P0144-lacZ* positive cells to more distal TC areas, especially in Tr2-5 (**B-C**, arrows). In *grn* mutants, expression of the P-fate markers *upd* (**D**) or *P0144-lacZ* (**E**) disappears while *arm-gal4*-mediated Grn overexpression restores *P0144-lacZ* expression in the proximal cells (**F**). (**G**) Scatter plots of the Trh-positive cell numbers in the whole Tr5 or in the SB/TC subregion of the indicated genotypes at stage 14 (mean ± SEM, N = 5; all p-values were calculated by Student's t-test). The cell number is decreased by around 10 cells in *grn* mutants in Tr5 and in the SB/TC subregion. A source file is supplied in *Figure 4—source data 1*. (**H**) Scatter plots of the number of metameres with any number of *P0144-lacZ*-positive cells of the indicated genotypes at stages15/16 (NS: not significant by Student's t-test). Note that grn mutants occasionally possess a few *P0144-lacZ* cells in anterior metameres. A source file is supplied in *Figure 4—source data 2*. (**I**) summarizes the two functions of *grn* in P-fate promotion. Scale bar: 50 μm. SB, spiracular branch; SEM, standard error of the mean, TC, transverse connectives.

The following source data and figure supplement are available for figure 4:

**Source data 1.** Source data for *Figure 4G*.

**Source data 2.** Source data for *Figure 4H*.

**Figure supplement 1.** Regulation and function of *grn* expression.

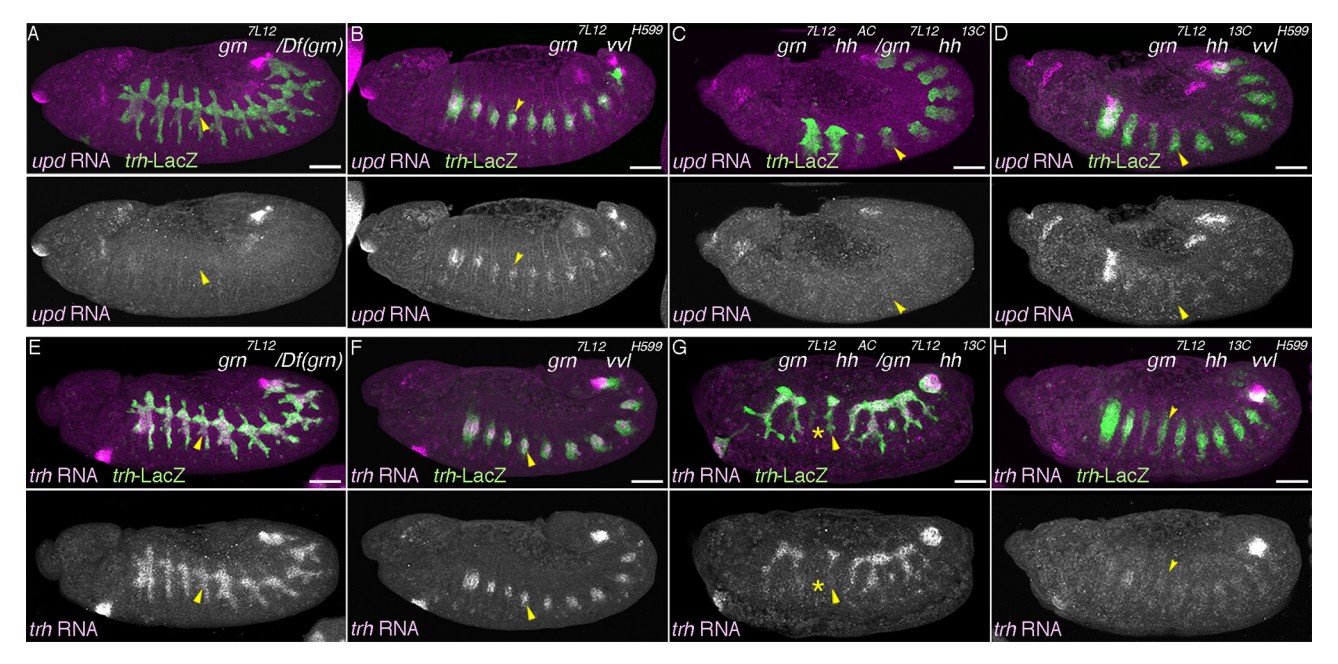

**Figure 5.** Genetic interactions of *grn, hh,* and *vvl* Expression of the P-fate marker *upd* (**A–D**, stages 12-13) or the pan-tracheal marker *trh* (**E–H**, stages 12-13) are shown. Loss of *upd* expression in *grn* mutants (**A**) is partially reversed by *vvl* mutation (**B**, arrowheads) but not by *hh* mutation (**C**). In the absence of all, *grn, hh,* and *vvl, upd* expression is virtually lost (**D**). This accompanies severe reduction of *trh* expression (**H**, arrowheads), compared to either *grn* (**E**), *grn vvl* (**F**), or *grn hh* double mutants (**G**). Note the *trh-lacZ* expression in **H**, reflecting initial *trh* expression at the primordia stage. Also, expression of *trh* and *upd* in Tr1 is often detected at stage 12. Asterisks in **G** marks loss of *trh* expression in Tr3. Scale bar: 50 μm.

The following figure supplements are available for figure 5:

**Figure supplement 1.** Genetic analysis of *grn, hh, vvl,* and RTK components.

**Figure supplement 2.** Phenotypes caused by simultaneous loss of the D- and P-factors.

detected weak restoration of the P-fate markers *upd* and *P0144-lacZ* (***Figure 5B***, ***Figure 5—figure supplement 1F, K***). These results may suggest the presence of another factor in the distal area that promotes the P-fate when *grn* and *vvl* are inactivated. Alternatively, the default airway cell fate is the proximal and multipotent cell fate (the P-fate) and an important aspect of the *grn* functions in P-fate selection/promotion maybe to antagonize the D-factor *vvl*. In the absence of both the D- and P-factors (***Figure 5H***, ***Figure 5—figure supplement 2A–B***), *trh* expression initiates at stage 10 but becomes drastically reduced by stage 12/13 (***Figure 5E–H***). In these mutants (***Figure 5D***, ***Figure 5—figure supplement 1D, H, M***, ***Figure 5—figure supplement 2A–B***), the expression of both proximal and distal markers is almost completely eliminated (***Figure 5A–D***. ***Figure 5—figure supplement 1A–N***). We conclude that the regulators of the D- and P-fate cooperatively promote and maintain the airway cell identity defined by *trh* expression. Below, we first describe roles of the embryo axis determinants on P/D-fate selection and then proceed to dissect the function of D-factors by gain-of-function approaches.

## Wg/WNT and Dpp/BMP signaling converge on the P/D-fate selection

A key aspect of the P/D-fate selection in the *Drosophila* airways is the separation of the *trh* positive airway primordia into a D-fate circle (*vvl, rho* and *btl* positive) surrounded by a concentric P-fate ring (*grn* and *upd* positive). To investigate the origins of this rough setup, we examined the effects of Wg/WNT and Dpp/BMP (***Figure 1B***), two major axis determinants along the AP or DV axis, respectively (***DiNardo et al., 1994***; ***Wharton et al., 1993***).

In *wg/WNT* mutants, expression of the D-fate markers *vvl* (*de Celis et al., 1995*), mab2A12 (*Lli-margas, 2000*), and *btl* expands along the AP axis to cover most, if not all lateral parts of the embryos as a single band (*Figure 6A–B, I*, *Figure 6—figure supplement 1B*). On the other hand, expression of the P-fate markers *grn, upd,* and *P0144-lacZ* also expands along the AP axis, but this time, approximately forms two narrow stripes, dorsal and ventral, sandwiching the D-fate (*Figure 6C, I*, *Figure 6—figure supplement 1A–B*). These results suggest that Wg/WNT confines both the D- and P-fates along the AP axis (*Figure 6I*).

The dorsal and ventral stripes of the proximal markers surrounding the expanded airways in *wg/WNT* mutants suggest that there are cues driving the P/D fate selection along the DV axis. Dpp/BMP presents a top candidate for such a signal (*Isaac and Andrew, 1996*; *Wilk et al., 1996*). In the absence of *dpp*, *trh* expression expands to the dorsal midline (*Figure 6—figure supplement 2A, D*) (*Isaac and Andrew, 1996*) and the cells initiate *btl* expression (*Figure 6—figure supplement 2B*). However, neither the D-fate nor the P-fate is established (*Figure 6—figure supplement 2A–E*), suggesting that Dpp also functions as a positive regulator of early airway development, in addition to its repressive effect on the earliest *trh* expression. In *dpp* hypomorphs (*Figure 6D–F*, *Figure 6—figure supplement 2F–J*), *trh* expression is expanded to the dorsal midline and P/D specification is confined near the dorsal midline, depending on the allele severities or the residual Dpp/BMP activity in the hypomorphic conditions. In the dorsally expanded primordia of *dpp$^{hr92}$* homozygotes, expression of the D-factors *vvl* and *btl* is detected as stripes straddling the dorsal midline (*Figure 6D–E*), and P-fate marker expression typically encircles the entire airway primordium except for its anterior border (*Figure 6F, I*, *Figure 6—figure supplement 2J*). The loss of the P-fate markers at the anterior border in *dpp$^{hr92}$* homozygotes could be due to the activity of the D-fate inducer Hh, which is expressed just anterior to the airway primordia (*Figure 1B*) (*Glazer and Shilo, 2001*). Consistent with the Hh-dependent anterior bias of the D-fate selection, the expression of the dorsal-distal marker *h-lacZ*, is lost in *hh* mutant (*Figure 6—figure supplement 3A–B*). Similarly, expression of *h-lacZ* is lost in *rho* and *pnt* mutants but not in *btl* mutants (*Figure 6—figure supplement 3A, C, D*). On the other hand, overactivation of Hh signaling in *wg/WNT* mutants expands *h* expression along the AP axis (*Figure 6—figure supplement 1N–Q*). Collectively, this analysis suggests that the graded Dpp/BMP activity controls the P/D fate selection at the dorsal border of the airway primordia. Wg/WNT signals impinge on the P/D-fate selection along the AP axis, while Hh modulates P/D differentiation from the anterior border. Consistent with the predominant roles of Dpp/BMP and Wg/WNT in airway primordia specification, in *Df(wg) dpp$^{hr92}$* double mutants, the whole dorsal area of the trunk/abdominal parts of the embryo takes the airway cell fate (*Figure 6G, I*). In this situation, the dorsal part of the airway primordia takes the D-fate and only the ventral periphery takes the P fate (*Figure 6H, I*), supporting our model that Dpp/BMP and Wg/WNT are key regulators of the P/D fate selection.

Several other cues including inducers of the airway fate (*Brown et al., 2001*) are expected to contribute to the P/D fate selection. At the ventral border for example, apart from the dorso-ventral gradient of Dpp/BMP, the ventro-dorsal gradient of *dEGFR* signaling (*Gabay et al., 1997*; *Golembo et al., 1996*; *Raz and Shilo, 1993*) and another cue originating from the ventro-lateral gradient of the TF Dorsal (*Anderson et al., 1985a*; *Roth et al., 1991*; *Zhao and Skeath, 2002*) may orient the P/D fate selection. The exact mechanisms by which Wg/WNT, Dpp/BMP, and these other cues impinge on P/D fate selection await future research and the development of appropriate cell-specific gene inactivation methods.

## Overactivation of the D-fate determinants abrogates the P-fate selection

In order to investigate how the P- and D-fates are balanced during airway cell differentiation, we analyzed the effects of overactivation of the D-fate determinants, namely RTK signaling, *hh* signaling and *vvl*. *argos* (*aos*) is a secreted negative feedback regulator of *dEGFR* signaling (*Schweitzer et al., 1995a*) while *anterior open* (*aop*) encodes an ETS TF that antagonizes RTK signaling (*Rebay and Rubin, 1995*). Upon activation of RTK signaling, *aos* is induced, while Aop is excluded from the nucleus and becomes degraded. In the airway primordia, *aos* expression gradually expands from the center to the periphery (*Figure 7—figure supplement 1A–E*). At early stage 12, both *aos* and Aop are preferentially detected in the proximal airways (*Figure 7—figure supplement 1E–F*). In *aop* mutants, the number of cells expressing P-fate markers is very much reduced (*Figure 7A*, *Figure 7—*

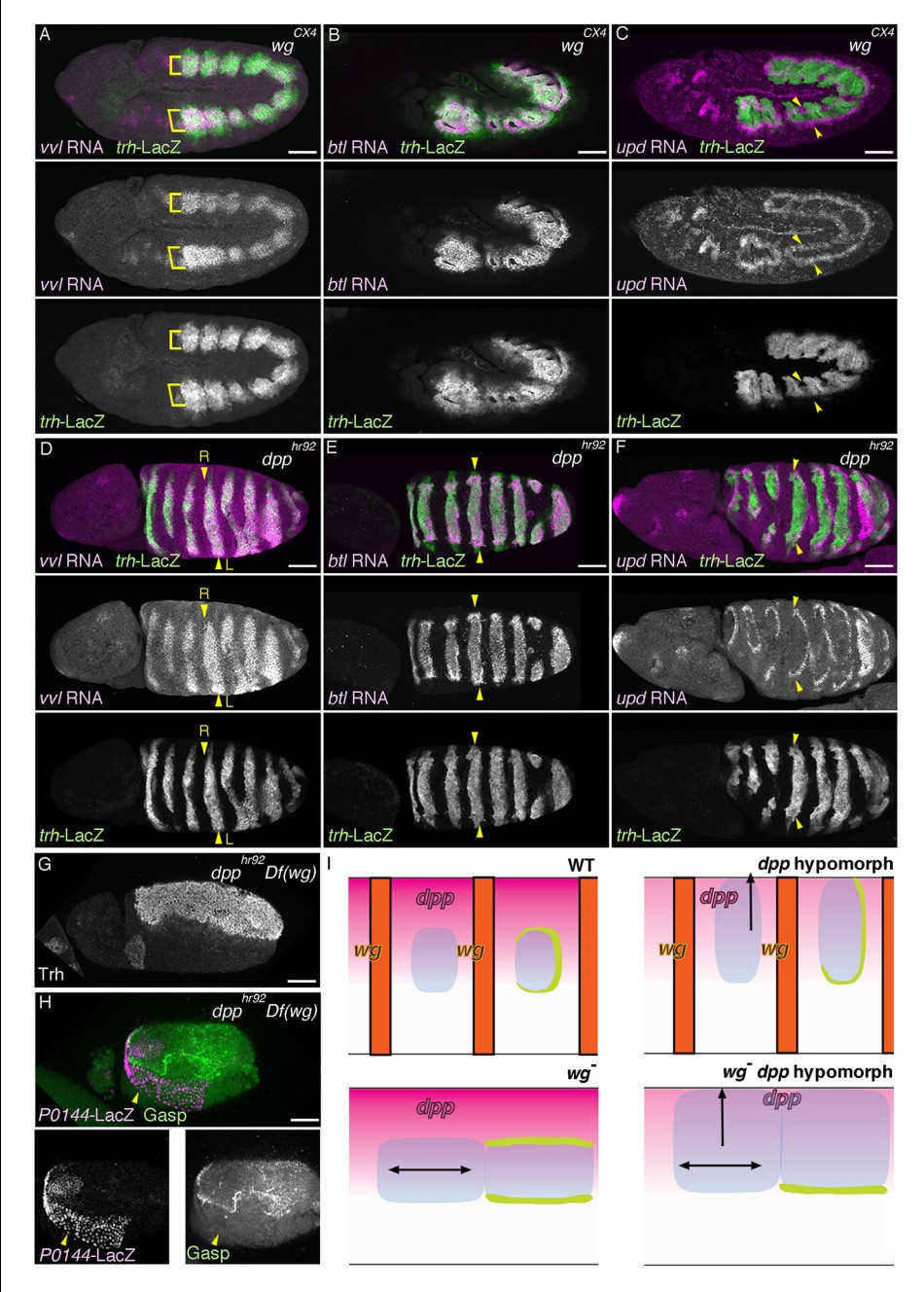

**Figure 6.** Regulation of the P/D fate selection by Wg/WNT and Dpp/BMP. Expression regions of the D-fate markers (*vvl* in **A**, **D**, *btl* in **B**, **E** or Gasp in **H**) or the P-fate markers (*upd* in **C**, **F** or *P0144-lacZ* in **H**) are shown for *wg* mutants (**A–C**), *dpp* hypomorph (*dpp$^{hr92}$*) (**D–F**) or *dpp$^{hr92}$ wg* double mutants (**H**). (**D-F**) are dorsal views where both left (**L**) and right (**R**) sides are seen. In *wg* mutants, expression domains of the D-fate markers expand along the AP axis (**A,B**), and form a single lateral band (bracket) with occasional interruptions before the initiation of primordia invagination (**A**, stage 10). After invagination (**B,C**, stage 12), some D-fate cells remain exposed at the embryo surface to form a lateral band (**B**). The P-fate markers form a dorsal and a lateral band of expression at the ectoderm (**C**, arrowheads). In *dpp$^{hr92}$* mutants (**D-F**), the airway primordia expand to the dorsal midline (**D**, stage 10). Concomitantly, both the D-fate (**D,E**) and the P-fate (**F**) expand to the dorsal midline. Note that the P-fate encircles the D-fate (arrowheads) and that the anterior cells often do not express the P-fate marker *upd* in (**F**, stage 11/12), probably due to the distalization by Hh that is expressed in the anterior border of each primordium. In *dpp$^{hr92}$ Df(wg)* double mutants (**G,H**), the whole dorsal area of the trunk/abdominal body parts become Trh positive at stage 11 (**G**). At stage 14/15 (**H**), the P-fate (arrowheads) is specified abutting the D-fate only at the

*Figure 6 continued on next page*

*Figure 6 continued*

ventral edge. Note that in the posterior part, the P-fate is not established as it is not established in the wild type. Scale bar: 50 µm. (I) Interpretation of data in each genotype regarding specification of the airway field (left parts) and P/D-fate selection (right parts). Expression domains of *dpp* and wg are colored pink and orange, respectively. In *wg* mutants, the airway field (light blue) expands along the AP axis (arrows) and the P-fate is established approximately as a dorsal and a ventral stripes (light green). In *dpp* hypomorphs (*dpp$^{hr92}$*), the airway field expands dorsally to the dorsal midline and the P-fate is established in the periphery except for the anterior and dorsal margin. In double mutants of *wg* and *dpp* hypomorph, the airway field expands both dorsally and along the AP axis, and the P-fate is only established in a ventral stripe.

The following figure supplements are available for figure 6:

**Figure supplement 1.** Role of *wg* in P/D fate selection.

**Figure supplement 2.** Roles of *dpp* in P/D fate selection.

**Figure supplement 3.** Roles of *D-factors* in *h* expression.

---

*figure supplement 2A*). Moreover, compared to either of the single mutants (*Figure 7B*, *Figure 7— figure supplement 2A*), in *aop aos* double mutants, expression of the P-fate marker *P0144-lacZ* is virtually abolished (*Figure 7C*, *Figure 7—figure supplement 2A*). Similar phenotypes are observed upon tracheal overexpression of *Ras$^{V12}$* (*Figure 7D*) or *s-spi* (*Figure 7E*) with *btl-gal4* or *trh-gal4* in an *aos* mutant background. In double mutants of *aos* and *Gap1*, a GTPase-activating protein for *Ras85D* (*Gaul et al., 1992*), expression of the P-fate markers *upd* and *grn* is variably reduced or abolished (*Figure 7G,H*). The expression of another P fate marker *P0144-lacZ* is also lost (*Figure 7F*, *Figure 7—figure supplement 2A–B*). Concomitantly, the distal mAb2A12/Gasp marker staining appears on the embryo outer surface (*Figure 7E–F*). These results suggest that *aos, Gap1,* and *aop* are parts of a negative regulatory mechanism balancing the D-fate inducing activity of RTK signaling to assure P-fate selection.

The loss of *P0144-lacZ* expression in *aos Gap1* double mutants is partially suppressed by overexpression of the P fate determinant Grn (*Figure 7I*, *Figure 7—figure supplement 2A*). Reversion of the *aos Gap1* mutant phenotypes is also detected by introducing *vvl, hh, rho, Ras85D* (*Simon et al., 1991*) or *pnt* mutations but not by *btl* or by *single-minded (sim)* mutations (*Figure 7—figure supplement 2A–I*). *sim* governs the ventral midline cell fate, which is one source of dEGFR signaling before stage 10 (*Golembo et al., 1996*; *Mayer and Nusslein-Volhard, 1988*). Thus, these results illustrate the essential roles of *vvl, hh,* and RTK signaling components as D-fate determinants and suggest that neither the CNS midline nor *btl/dFGFR* are involved in the RTK overactivation observed in the *aos Gap1* double mutants. Similarly, *arm-gal4* mediated, ubiquitous overexpression of secreted active *spitz (s-spi)* results in loss of the P-fate marker *P0144-lacZ* (*Figure 7—figure supplement 2J– K*). This defect is also suppressed by mutations of *vvl, pnt,* or *Ras85D* but not by *btl, rho,* or *hh* mutations (*Figure 7—figure supplement 2J–P*). This suggests that *vvl* has an additional essential role in dEGFR signaling other than facilitating active dEGFR ligand production. In contrast to *s-spi, bnl* overexpression does not abolish *P0144-lacZ* expression (*Figure 7—figure supplement 1Q*). However, simultaneous overexpression of an activated form of *btl/dFGFR* and its downstream mediator *downstream of FGFR (dof)* (*Imam et al., 1999*; *Michelson et al., 1998*; *Vincent et al., 1998*) reduced *P0144-lacZ* expression (*Figure 7—figure supplement 1R*) suggesting that *dEGFR* and *btl/ dFGFR* share the downstream signaling pathway for P/D fate selection.

Overactivation of *hh* signaling, either by mutation of the inhibitory receptor *ptc* (*Chen and Struhl, 1996*; *Ingham et al., 1991*) or by *arm-gal4*-mediated *hh* overexpression, reduces the number of cells expressing P-fate markers (*Figure 7J–K*). Although overactivation of Hh signaling is expected to increase the expression of Wg/WNT (*DiNardo et al., 1994*), a negative regulator of *trh* expression (*Wilk et al., 1996*), reduction of the P-fate cell number upon *hh* overactivation still occurs in *wg/ WNT* mutant backgrounds (*Figure 7M–N*). Moreover, driving a repressor form of Cubitus interruptus (Ci$^{rep}$), a mediator TF of Hh signaling (*Hepker et al., 1997*; *Methot and Basler, 2001*) at the dorsal ectoderm with *salm-gal4* in *ptc wg* mutants locally increases the P-fate cell number in the dorsal side

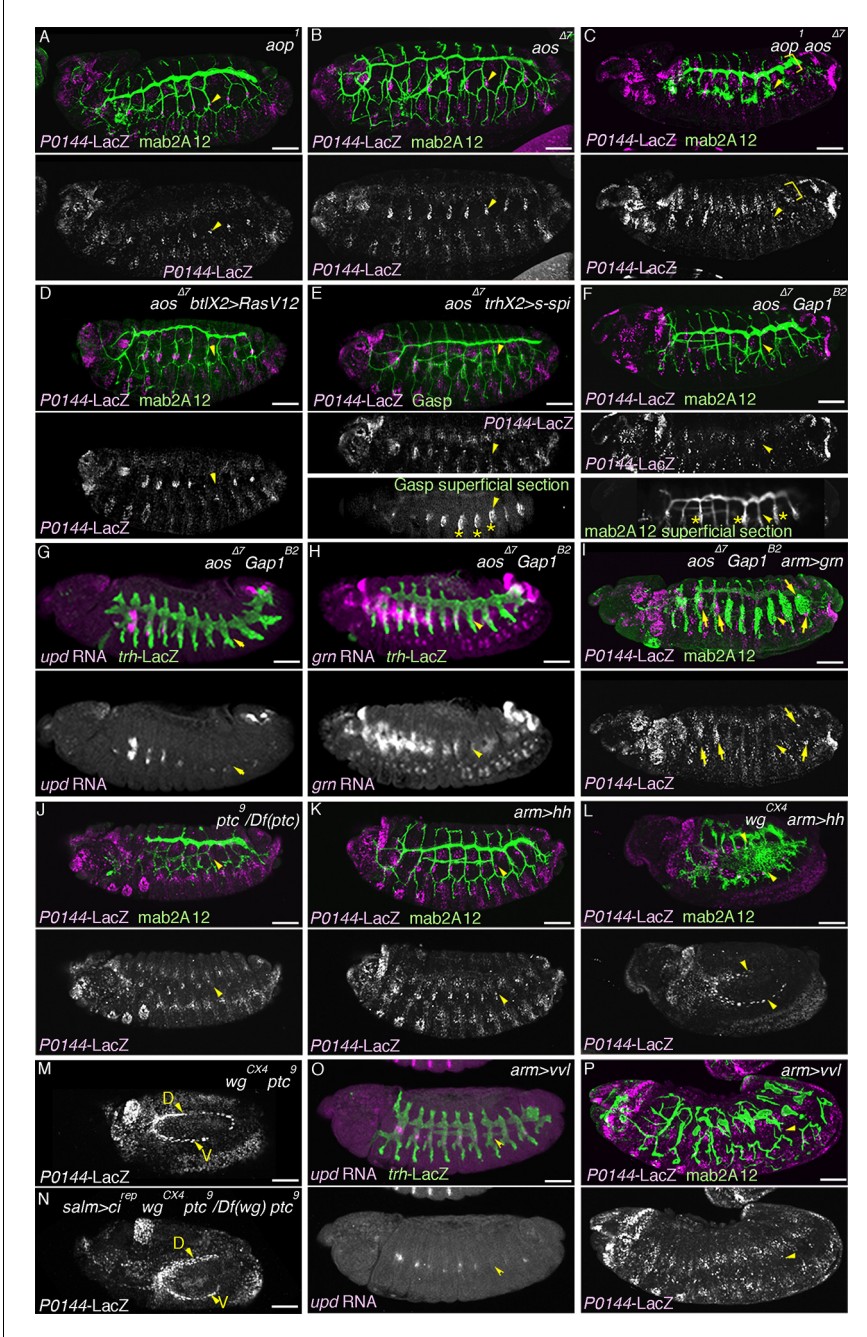

**Figure 7.** Overactivation of the D-fate determinants abrogates the P-fate specification. (A–I) Effects of RTK signaling overactivation. Proximal areas are marked by arrowheads in Tr7. In *aop* mutants (A), expression of the P-fate marker *P0144-lacZ* is variably decreased or lost. Compared to either single mutant (A,B), in *aop aos* double mutants (C), *P0144-lacZ* expression is virtually abolished. Note that the epidermal signals in C (brackets) are from other lineages. *aos* mutation, combined with $Ras^{V12}$ overexpression by *btl-gal4* (D), *s-spi* overexpression by *trh-gal4* (E) or *Gap1* mutation (F) abolish *P0144-lacZ* expression. Note that in E and F, after enhancement of mab2A12/Gasp signals, the epidermal surface staining is evident (asterisks). In *aos Gap1* double mutants, expression of the other P-fate markers (*upd* in G and *grn* in H) is variably reduced or lost, while *grn* overexpression (I) partially restores *P0144-lacZ* expression to the proximal areas of *aos Gap1* double mutants (arrows). (J–P) Effects of overactivation of *hh* or *vvl*. Overactivation of *hh* signaling by *ptc* mutation (J,M) or *hh* overexpression (K,L), either in the wild-type background (J,K) or *wg* mutant background (M,L) reduces the *P0144-lacZ*-positive cells. *salm-gal4* mediated overexpression $ci^{rep}$ increases the area of *P0144-lacZ*-positive P-fate cells in the dorsal stripe (N). The dorsal and the ventral stripes are marked with D and V in (M,N). *vvl* overexpression reduces or abolishes

*Figure 7 continued*

the expression region of the P-fate markers *upd* (**O**) or *P0144-lacZ* (**P**). Scale bar: 50 μm. RTK, receptor tyrosine kinase.

The following source data and figure supplements are available for figure 7:

**Source data 1.** Source data for *Figure 7—figure supplement 2A*.

**Source data 2.** Source data for *Figure 7—figure supplement 2J*.

**Figure supplement 1.** Expression of *aos* and Aop.

**Figure supplement 2.** Requirement of the overactivated RTK signaling in the abrogation of the P-fate selection.

**Figure supplement 3.** Effects of *vvl* or *s-spi* overexpression in *wg/WNT* mutant embryos.

---

(*Figure 7N*). These results imply that overactivation of *hh* signaling autonomously represses the P-fate selection independent of its effect on *wg/WNT*.

Overexpression of *vvl* with *arm-gal4,* like *s-spi* overexpression reduces or abolishes the expression of the P-fate markers, *upd* or *P0144-lacZ* (*Figure 7O–P*), and this occurs independently of the presence of wg/WNT (*Figure 7—figure supplement 3A–C*). Collectively, the analysis indicates that expansion of the D-fate-inducing activities of RTKs, Hh or Vvl is deleterious to the P-fate selection and that the negative regulators of RTK signaling (*aos, aop* or *Gap1*) and *hh* signaling (*ptc*) ensure the selection or maintenance of the P-fate.

## Differential competence of the primary branches and the TC

The hitherto analysis indicates that *aos* sensitizes the circuit of P/D-fate selection. We noticed that crossing of a weaker driving strain of *UAS-Ras^{V12}* with *arm-gal4* causes loss of *P0144-lacZ* expression only in the *aos* mutant background while the stronger *UAS-Ras^{V12}*strain is sufficient to eliminate *P0144-lacZ* expression on its own (*Figure 8—figure supplement 1A–D*). Using the weaker *UAS-Ras^{V12}* strain, we demonstrate that within the D-fate group, the cells of the TC and the remaining primary branches have differential competence (*Figure 8—figure supplement 1E*). In wild-type embryos, Kni and Knrl are induced in a subset of the distal primary branches (DB, LT and GB) in response to Dpp/BMP (*Chen et al., 1998*; *Vincent et al., 1997*). *arm-gal4*-mediated overexpression of Dpp/BMP or the active form of its co-receptor tkv^{QD} (*Nellen et al., 1996*) is sufficient to induce ectopic Kni in additional branches (*Figure 8A,B*) (*Vincent et al., 1997*). However, Kni levels in the TC cells is weaker compared to the more distal cells, suggesting that TC cells are less competent in inducing the D-fate marker Kni in response to Dpp/BMP signaling. When both *UAS-dpp* and the weaker line of *UAS-Ras^{V12}* are simultaneously driven by *arm-gal4*, kni expression becomes homogenously induced in both the TC cells and the more distal cells (*Figure 8C*). This suggests that graded activity of the D-fate inducers, like RTKs may act to generate three different cell states along the PD axis of the airways. The most distal part (the primary branches) is established in response to the highest activity, the intermediate domain (TC) requires weaker activity, whereas the level of the D-factors needs to be low in the most proximal part (*Figure 8—figure supplement 1E*). Slight expansion of the P-fate only in the TC region upon Grn overexpression (*Figure 4A–C*) may also reflect this differential competence of TC and the remaining primary branches.

The early development of the airway primordia includes two essential aspects: first, the selection and maintenance of the airway field, and second, the selection of the P/D-fates, both of which originate from the AP and DV axis determinants of the embryo (*Figure 9*). More specifically, we identified a positive role of Dpp/BMP and the P/D fate factors in the establishment of the airway field. In the absence of Dpp/BMP or both P/D fate regulators, the airway field is lost. The latter phenotype could be explained in the following way in terms of P/D fate selection. Without the D-factors, the airway field is expected to uniformly take the P-fate that is promoted by *grn*. In this situation, *grn* becomes indispensable for maintenance of *trh* expression in the main airways. However, Wg/WNT, Hh, Dpp/BMP and other cues generate a centrally enriched expression field of the D-fate

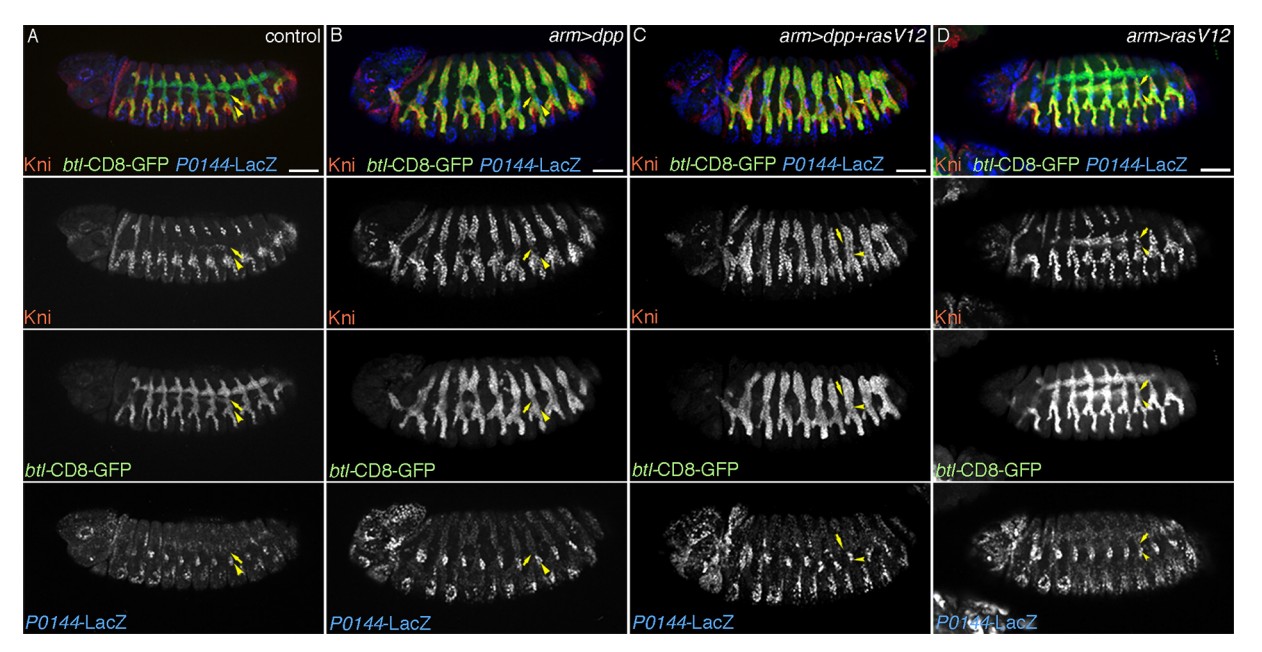

**Figure 8.** TC cells may define an intermediate fate between the more distal primary branches and the SB. Expression of Kni is shown together with the D-fate marker *btl-CD8GFP* and the P-fate marker *P0144-lacZ* upon overexpression of *dpp/BMP* and/or *Ras*^V12^. Arrowheads and arrows mark the P-cells and TC in Tr7, respectively. Compared to the control, *dpp/BMP* overexpression induces Kni expression in all the D-cells. But the Kni expression level is weaker in TC. The differential Kni expression levels in the primary branches and TC (**B**) become equalized by additional overexpression of *Ras*^V12^ (**C**). Upon *Ras*^V12^ overexpression alone (**D**), Kni expression becomes detected along the TC. Note that in all cases, *P0144-lacZ*-positive P-cells do not express Kni. Scale bar: 50 µm. SB, spiracular branch; TC, transverse connectives.

The following source data and figure supplement are available for figure 8:

**Source data 1.** Source data for *Figure 8—figure supplement 1D*.

**Figure supplement 1.** Characterization of the effects of the weaker and the stronger *UAS-Ras*^V12^ lines.

determinants *vvl, rho,* and *btl*, which then cooperate to repress the P fate, leaving the *grn-trh* regulation operative only in the P-region. Restoration of the P-fate in *grn vvl* double mutants suggests that the proximal multipotent fate is the default state for airway cells or that there is another factor that promotes the P-fate in the absence of *grn* and *vvl*.

Intriguingly, the *Drosophila* legs and the trachea are supposed to have evolved form common ancestor appendages (*Franch-Marro et al., 2006*). The proximo-distal patterning mechanisms of these two organs are similar in that RTK activation distalizes the field (*Campbell, 2002*; *Galindo et al., 2002*) while initiation of the distal identity is conferred by the same set of signaling molecules (Dpp/BMP, Wg/WNT, and Hh) from the outside (for the airways) or the inside (for the legs) (*Estella et al., 2012*). The proximal location of multipotent cells in the *Drosophila* airway tree resembles the conspicuous proximal localization of multipotent tracheal basal cells in the mouse lung (*Rock et al., 2009*; *Rock et al., 2010*) Given the prominent roles of FGFR signaling in branching morphogenesis in both flies (*Sutherland et al., 1996*) and mice (*Min et al., 1998*; *Sekine et al., 1999*), the identification of a genetic mechanism for P- and D-fate selection in flies may aid future studies aiming to identify the mechanisms that spatially confine multipotent cell selection in the embryonic vertebrate lung.

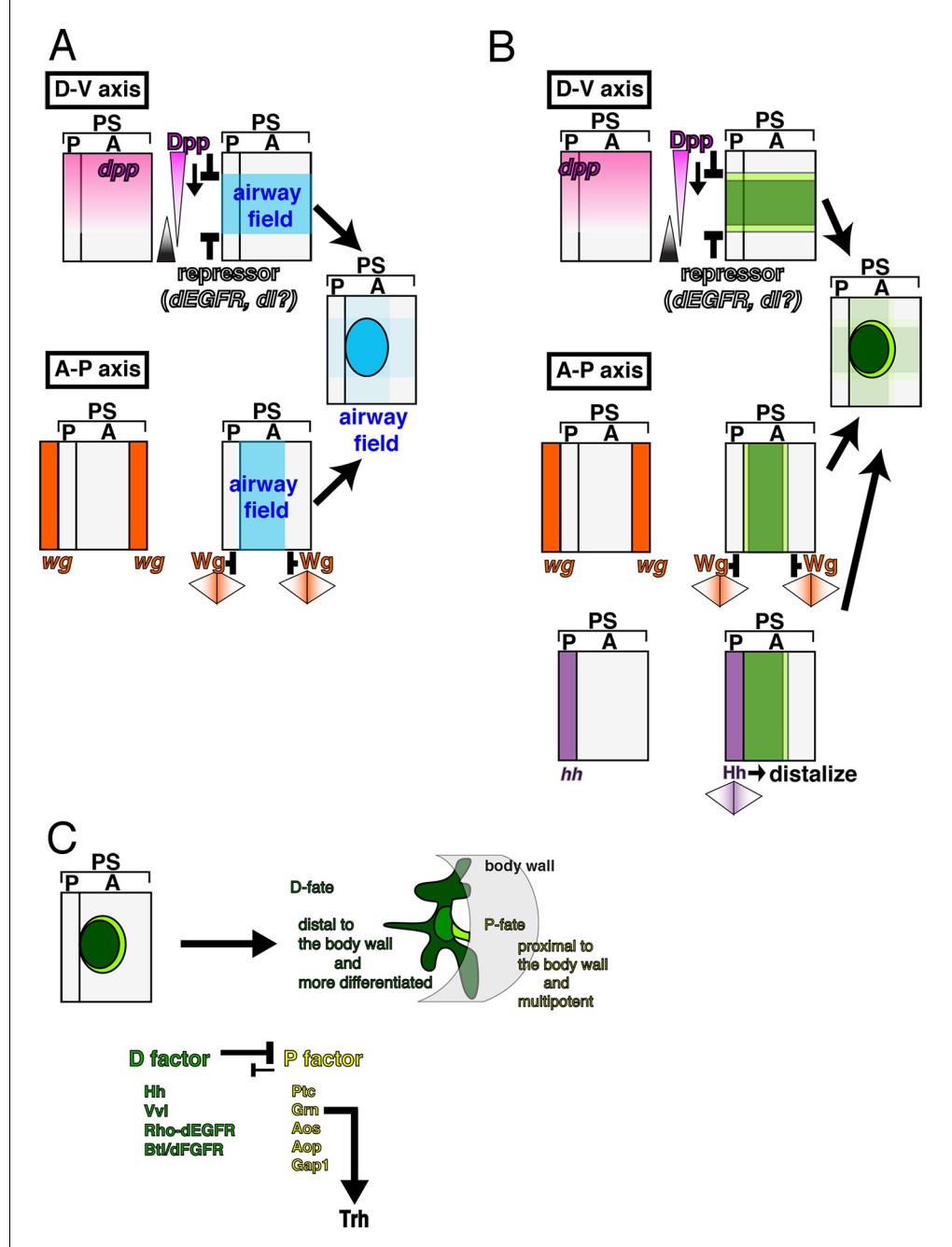

**Figure 9.** Genetic circuits explaining the early development of the *Drosophila* airways. (**A**) Establishment of the airway field is regulated by Dpp/BMP and Wg/WNT. In each metameric unit of the *Drosophila* embryo (parasegment, PS) (***Sanson, 2001***), an airway primordium is specified just posterior to the *hh* expression domain (a posterior compartment of a segment, P). This is combinatorially controlled along the DV and AP axis. Dpp/BMP expressed in the dorsal region functions as both a repressor (***Isaac and Andrew, 1996***; ***Wilk et al., 1996***) and an activator of the airway field (this study) while Wg/WNT functions as a repressor along the AP axis. (**B**) Initiation of the P/D-fate selection by Dpp/BMP, Wg/WNT, and Hh. Each airway primordium is roughly subdivided into two regions, anterior-central (D-fate, dark green) and the peripheral (P-fate, light green). The patterning cues along the AP and DV embryo axis roughly set up this radial patterning. Along the DV axis, Dpp/BMP expressed in the dorsal region functions to discriminate the P/D-fates, at least at the dorsal edge. Dpp/BMP may also discriminate the P/D-fates at the ventral edge. Alternatively, ventral cues dependent on the Dorsal TF gradient may discriminate the P/D-fates at the ventral edge. Along the AP axis, Wg/WNT expressed in transverse stripes may discriminate the P/D-fates at each edge. Hh from the anterior border stimulates the D-fate. (**C**) Establishing the P/D-fates by the P/D-

*Figure 9 continued*

factors. The ordered Invagination of the primordia converts the centro-peripheral patterning to the proximo-distal organization along the airway tree. By completion of invagination, D-factors establish the D-fate at the expense of the P-fate. Within the D-fate, there are two groups of cells having different responsiveness to overexpression of Dpp/BMP or Grn, e.g., TC (green) and the primary branches (dark green). See text for details. AP, anterior-posterior; DV, dorso-ventral.

## Materials and methods

### Fly genetics

Flies kept over balancer chromosomes (*Lindsley et al., 1992*) were grown in standard medium. We obtained the appropriate genotypes by standard genetic crosses. For overexpression of genes, we used the Gal4/UAS system (*Brand and Perrimon, 1993*). Germline clones of *Ras85D$^{c40b}$* mutants were made by FLP-DFS technique (*Chou and Perrimon, 1996*; *Hou et al., 1995*). Mutant embryos were identified by the expression of *twi-lacZ, ftz-lacZ, hb-lacZ*, Ubx-lacZ, or GMR-dfd-GFP constructs inserted on balancer chromosomes. We identified mutants harboring *dpp* mutations by selecting embryos with previously reported phenotypes in the embryo DV patterning. For collection of large numbers of virgins, we used a Y chromosome harboring *hs-hid* construct developed by R. Lehmann and M. Van Doren (*Starz-Gaiano et al., 2001*). See Flybase (*St Pierre et al., 2014*) for details of strains described below.

Mutant strains; *AbdB$^{M1}$* (a gift from I. Lohmann) (*Lohmann et al., 2002*), *btl$^{Δoh10}$* and *btl$^{Δoh24-1}$*(*Ohshiro and Saigo, 1997*), *Df(os)1A* (a gift from D. Harrison) (*Harrison et al., 1998*), *Gap1$^{B2}$* (a gift from N. Perrimon) (*Hou et al., 1995*), *grn$^{7L12}$* (a gift from J. Hombria) (*Brown and Castelli-Gair Hombria, 2000*), *hh$^{13C}$* (*Hosono et al., 2003*), *Ras85D$^{Δc40b}$* (a gift from N. Perrimon and C. A. Berg) (*Hou et al., 1995*; *Schnorr and Berg, 1996*), *rho$^{Δ38}$* (a gift from D. Andrew) (*Bradley and Andrew, 2001*), *rho$^{7M}$* (a gift from J. Skeath) (*Skeath, 1998*), *top$^{f24}$* (a gift from K. Moses) (*Kumar et al., 1998*) *ut$^{H599}$* = *vvl$^{H599}$* (a gift from A. Salzberg) (*Inbal et al., 2003*). *arm$^4$* was obtained from National Institute of Genetics (NIG), Mishima, Japan. *aop$^1$*, *aos$^{Δ7}$*, *bnl$^{P1}$*, *Df(3R)Dl-BX12 as Df(bnl)*, *dpp$^{hr92}$*, *dpp$^{H46}$*, *pnt$^{Δ88}$*, *Df(H99)*, *hh$^{AC}$*, *pnt$^2$*, *ptc$^9$*, *Df(2R)Exel7098 as Df(ptc)*, *sim$^2$*, *top$^{f2}$*, *trh$^2$*, *Df(3L)Exel6109 as Df(vvl)*, *wg$^{CX4}$*, *Df(2L)Exel6017 as Df(wg)* were obtained from Bloomington stock center (BDSC), Indiana.

Enhancer trap strains; *1-eve-1 as trh-lacZ* (a gift from N. Perrimon) (*Perrimon et al., 1991*) and *P0144-lacZ* (a gift from W. Janning, Flyview), *pnt-lacZ* (a gift from M. Krasnow) (*Samakovlis et al., 1996a*). *h-lacZ* was obtained from BDSC.

Enhancer reporter strains; *btl-CD8-GFP* (a gift from M. Sato) (*Sato and Kornberg, 2002*), *kni-(dpp)-lacZ* (*Chen et al., 1998*), and *salm-TSE-lacZ* (*Kuhnlein and Schuh, 1996*) (gifts from R. Schuh).

Gal4 and UAS strains; *btl-gal4* and *trh66-gal4* on sencond and third chromosomes (gifts from S. Hayashi) (*Kondo and Hayashi, 2013*; *Shiga et al., 1996*), *salm-gal4* (a gift from M. Llimargas) (*Lli-margas, 2000*), *UAS-bnl* (a gift from M. Krasnow) (*Sutherland et al., 1996*), *UAS-btl/dFGFR* (R. Matsuda and K. Saigo), *UAS-ci$^{75}$* (a gift from S. Ishii) (*Dai et al., 2003*), *UAS-ci$^{rep}$* (a gift from A. Moore) (*Karim and Moore, 2011*), *UAS-DIAP* (a gift from E. Kuranaga and M. Miura) (*Kuranaga et al., 2002*), *UAS-dof* (a gift from M. Affolter) (*Vincent et al., 1998*), *UAS-dpp* (a gift from K. Basler) (*Ruberte et al., 1995*), *UAS-grn* (a gift from J. Hombria) (*Brown and Castelli-Gair Hombria, 2000*), *UAS-hh* (*Hosono et al., 2003*), *UAS-torso$^{4021}$-btl* (a gift from E. Hafen) (*Feldmann et al., 1999*), *UAS-Ras$^{V12}$* (gifts from G. M. Rubin) (*Karim and Rubin, 1998*), *UAS-sspi* (a gift from B. Shilo) (*Schweitzer et al., 1995b*), *UAS-vvl* and *UAS-vvl vvl$^{H599}$* (gifts from A. Salzberg) (*Inbal et al., 2003*). *arm-gal4, UAS-dEGFR* and *UAS-nGFP* were obtained from BDSC.

### In situ hybridization and immunostaining

Eggs were collected on apple/grape juice plates at 25°C. Embryos were bleached and fixed as previously described (*Patel, 1994*) for 15–30 min with a 1:1 mixture of heptane and a fix solution (3.7% formaldehyde, 0.1 M Hepes pH6.9, 2 mM MgSO$_4$). Embryos were dechorionated with methanol and incubated in 0.1% PBT supplemented with 0.5% BSA. Staging of embryos was done as previously described (*Campos-Ortega and Hartenstein, 1997*).

For immunostaining, the following primary antibodies were used. Guinea-pig anti-Gasp (1:1000) (*Tiklova et al., 2013*), Guinea-pig anti-Kni (1:300), (developed by J. Reinitz and distributed by Y. Hiromi, East Asian Segmentation Antibody Center, Mishima, Japan) (*Kosman et al., 1998*), rabbit anti-Trh (1:50). Mouse anti-Abd-B (1:10, donated by S. Celniker) (*Celniker et al., 1989*), mouse anti-Aop (1:10, donated by I. Rebay and G. M. Rubin) (*Rebay and Rubin, 1995*), mouse Dcad (1:10, donated by T. Uemura) (*Oda et al., 1994*), mouse mab2A12 (anti-Gasp) (1:5, donated by M. Krasnow, N. Patel and C. Goodman) (*Samakovlis et al., 1996b*; *Tiklova et al., 2013*) were obtained from Developmental Studies Hybridoma Bank (DSHB), Iowa. Commercially available antibodies were anti-LacZ (*E. coli.* β-Galactosidase) antibodies made in goat (1:500, Biogenesis, UK) or rabbit (1:1000, Cappel, Netherlands) and anti-GFP antibodies made in rabbit (1:500, JL-8 Clontech, Mountain View, CA) or mouse (1:1000, GFP20 Sigma, St. Louis, MO).

Donkey or goat biotin- or fluorescently labeled secondary antibodies made against the host species of primary antibodies were purchased from Jackson Laboratories, Sacramento, CA. Streptavidin coupled with AMCA, FITC, or Cy5 were used when necessary. For mab2A12 detection TSA amplification (PerkinElmer, Waltham, MA) was used. For detection of apoptosis, a TUNEL kit from Roche (Switzerland) was used.

Double fluorescent labeling with RNA probes and antibodies was carried out as described (*Goto and Hayashi, 1997*). The following cDNA clones were used to make hybridization probes; *btl* (*Ohshiro and Saigo, 1997*), *h* (a gift from D. Ish-Horowicz) (*Hooper et al., 1989*), *grn* (a gift from J. Hombria) (*Brown and Castelli-Gair Hombria, 2000*), *pnt* (a gift from C. Klambt) (*Klambt, 1993*), *salm* (a gift from R. Schuh) (*Kuhnlein and Schuh, 1996*), *upd* (a gift from D. Harrison) (*Harrison et al., 1998*), *vvl* (a gift from J. Casanova) (*Llimargas and Casanova, 1997*). DNA clones of *aos, rho* and *trh* were obtained from Drosophila Genomics Resource Center (DGRC), Indiana, USA.

Confocal images were taken by Bio-Rad (Hercules, CA) MRC1024, Olympus (Japan) Fluoview 1000 or Zeiss (Germany) LSM780. Images of controls and mutants taken by the same confocal microscopes were used for comparison. Images were processed by ImageJ and figures were prepared with Adobe Photoshop and Illustrator.

## Acknowledgements

We thank the members of the fly community who isolated, characterized or distributed fly strains, antibodies or DNA clones. Especially, we thank M Affolter, D Andrew, K Basler, CA Berg, J Casanova, E Hafen, D Harrison, S Hayashi, Y Hiromi, J Hombria, D Ish-Horowicz, S Ishii, W Janning, C Klambt, M Krasnow, E Kuranaga, M Llimargas, M Miura, A Moore, K Moses, N Perrimon, GM Rubin, A Salzberg, M Sato, R Schuh, B Shilo, J Skeath, R Ueda, BDSC, DGRC and NIG for directly sharing fly strains and DNA clones. We thank Flybase for the Drosophila genomic resources. We thank the Stockholm University Imaging Facility and members of the Mannervik, Saigo, Samakovlis and Åström laboratories for support during the project, especially M Björk for fly service and V Tsarouhas for continuous support. Special thanks to Y Emori and F Ui-Tei for help in maintaining fly strains after the retirement of KS and to J Muhr for critical comments on the manuscript.

## Additional information

### Funding

| Funder | Author |
| --- | --- |
| Ministry of Education, Culture, Sports, Science, and Technology | Kaoru Saigo |
| Vetenskapsrådet | Christos Samakovlis |
| Cancerfonden | Christos Samakovlis |

The funders had no role in study design, data collection and interpretation, or the decision to submit the work for publication.

## Author contributions
RM, Conceived the project and designed the experiments, performed experiments, collected and analyzed data, drafted and wrote the manuscript; CH, Performed experiments, collected data; CS, Provided reagents and analysis tools, wrote the manuscript; KS, Provided reagents and analysis tools

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
