## [Decision Letter]

Thank you for submitting your work entitled "Multipotent versus differentiated cell fate selection in the developing *Drosophila* airways" for peer review at *eLife*. Your submission has been favorably evaluated by K VijayRaghavan (Senior editor, who also served as Reviewing editor), and two reviewers.

The following individual responsible for the peer review of your submission has agreed to reveal their identity: Bruce Edgar (peer reviewer).

The reviewers have discussed the reviews with one another and the Reviewing editor has drafted this decision to help you prepare a revised submission.

Summary:

This paper provides a careful, relatively comprehensive genetic analysis of signaling molecules and transcription factors that orchestrate cell specification during tracheal development in *Drosophila* embryos. The study focuses on two existing cell populations, which differ in their location and developmental potential: 1. P-fate precursors, a multipotent population located proximal to the body wall and 2. D-fate precursors, a more differentiated cell pool residing distal to the body wall. First, the authors define a set of markers to visualise P and D fate precursors: btl and mab2A12 to mark D-fate and P0144-enhancer trap and upd to stain for P-fate.

They continue to identify the underlying molecular networks controlling P and D fate selection: The transcription factor Grain (grn) promotes P-fate selection, whereas vvl stimulates D-fate decision. In addition, vvl together with Hh and RTK signalling (EGF and FGF) support D-fate at the expense of P-fate. More globally, they show that gradients of BMP, Hh and Wnt regulate the two cell populations to control distal-proximal organisation of distinct tracheal cell precursors.

The paper provides a lot of good quality data that supports most of its conclusions. This data is nearly all expression patterns in mutants, or in embryos with ubiquitous over- expression of one of the effector genes. Nevertheless, the interplay between different signalling cascades to coordinate of P-fate and D-fate selection is interesting, and will serve as a basis for further analysis of this developmental system. Important concerns do need to be addressed for acceptance and these are given below.

Essential revisions:

Overall this is a compelling piece of work, but there are many places where the manuscript would benefit from improved clarity. The paper suffers from trying to deliver too much data, and some of the material in the figures is redundant. It could be improved by streamlining the text, making the figures more compact and moving redundant data sets to supplemental locations (in general we would like to minimise the use of supplemental figures and have data in the main text, but do feel here that both clarity of figures and making all data available can be better reconciled in this paper by the use of supplemental figures.). So, fewer figure panels and clearer figures will bring much required clarity.

From the outset, the definition of D- and P-fate precursors is not clear enough, and this makes it difficult to follow the aims, experimental flow and the conclusions. Including a better schematic at the very beginning illustrating i) the organization and the development of the airway system on a cellular basis, and ii) the location and definition of P-and D-fate cells and their corresponding markers would be very beneficial. The schematic in Figure 1 is not very self-explanatory and the legend is cryptic. Similarly, the schematic figures illustrating the authors' conclusions (Figure 9) are complex and one reviewer found it difficult to follow and another would have like to see helpful aspects referred to earlier in the paper.

Some stainings especially the P-fate marker P0144-LacZ (e.g. Figure 1, Figure 2 lower panel) do not always seem to be specific and the background is very high, so that it is difficult to see what the authors claim. Virtually all the panels show whole embryos, and the magnification his not high enough to resolve individual cells. Higher magnification pictures of single segments would help to support the most important conclusions. In addition, since many of the effects shown are partial, quantifications would help to clearly define the impact of the different pathways on D-fate precursors.

Finally, virtually all of the genetic analysis is done with mutants that affect many non-airway cells as well as the airway cells, and so many of the effects shown may be indirect, non-cell autonomous effects. For instance the top and btl mutants enhance cell death in a broad swathe of ectodermal cells, and the hh mutants simply lack parts of segments that include airway precursor cells, probably because these cells were simply not generated much earlier in development. These issues might be addressed spatially and temporally targeting the suppression of gene activity to better resolve when and how the noted defects arise in the mutants. The targeted suppression of gene function by RNAi would be of value, particularly of genotypes where the effects of the mutants are likely to be very pleiotropic and there is a possibility that the effects on the trachea are secondary. This is particularly relevant to Figure 6.

Thus, while the study should prove to be useful, we think it requires some refinement before it is ready for publication.

Specific major comments:

1) Figure 1 shows the evolution of marker gene expression in the developing trachea. It would benefit from a clearer diagram indicating the markers that will be used for P and D fate in the subsequent figures.

2) The text is somewhat unclear on the relationship between vvl and RTK signalling and at times emphasises the co-operative effects, and at times the data that the RTK ligands are partly downstream of vvl (and Hh). Tightening the summary sentences would help.

3) Figure 4. By contrast, the authors show that P fate requires the transcription factor Grn. And that in the absence of the D fate, the Grn domain is expanded and cells acquire the default P fate. Is the D fate at all expanded in the grn mutants? It looks like it is (Figure 4 versus E), but this is not explicit in the text.

4) Figure 5. The epistasis experiments suggest that any residual vvl expression in the grn mutants (e.g. in the grn, Hh double mutants compared to the grn, vvl mutants) is sufficient to prevent P fate development. This strongly suggests that an important role of vvl is to inhibit P fate allowing the D fate to be specified. Indeed, that P is the default fate which must be suppressed by the D-promoting molecules is basically the authors' conclusion. However, the explanation for the effects (that there is substantial vvl expression in the hh mutants) is buried in Figure 3—figure supplement 1. Can the explanation in the text of Figure 5 be improved to make the manuscript easier to read? Given the epistasis experiments in Figure 5, and that P cells can be specified in the absence of grn (e.g. in the grn, vvl double mutants), I don't think it is appropriate to refer to Grn as a master-regulator of P fate in the previous section (Figure 4 description).

5) Figure 6. Analysis of Wg and Dpp mutants shows that P/D fate is strongly changed in these genotypes. This figure needs additional experiments to clarify the effects of Wg and Dpp. Is A-P and D-V patterning just compromised overall in these mutants, hence very strong effects on P and D fate in the trachea? Or, is there a direct Wg/Dpp signalling input into P and D fate? If RTK or Hh ligand expression (pathways the authors have shown to be required for P/D tracheal patterning) was examined at an earlier stage, would these be altered thus explaining the dramatic changes in the Wg/dpp mutants? An age-matched control shown within Figure 6 (even if repetitive with other figures, and yes, we see that we are contradicting ourselves as we previously asked that redundancy be avoided, but it this instance it will be useful) would assist the reader. (See comments on Figure 9 below). One way to address our concerns here would be to use temporal and tissue specific drivers with RNAi, particularly where concerns on pleiotropic effects can be argued to be of significant concern. Another way, not necessarily excluding this approach would be by live-imaging of phenotypes. In our editorial consultations we concluded that the latter is not a necessary requirement but leave it to the authors to best address this important concern in the manner(s) best suited.

6) Figure 7 supports the hypothesis from Figure 5 that D-fate determinants suppress the default P fate. Would Figure 7 be better following on from Figure 5?

Figure 7. It appears as if there is some residual P fate in this picture? Maybe these are cells of another lineage and it would benefit from some arrows, or maybe the text needs modifying?

7) Figure 8 shows that the various branches of the D part of the trachea respond differently to Dpp signalling, suggesting that the D factors may also affect later tracheal patterning.

8) Figure 9. This is a useful diagram but currently a bit complex. Could it be simplified? Perhaps the reader could be taken through the figure more gently by elaborating the legend? (This figure abrogates some earlier comments about Figure 6, though the text still needs to be clarified in that part). Are panels of the figure adapted from Sanson et al. 2001 just depicting their conclusions? In either case the panels should be appropriately referenced.

9) Are these phenotypes truly representative of 100% penetrance in every genotype used? Or is there some variation which should be made explicit in the manuscript?

10) The Introduction starts with a discussion of vertebrate pluripotency which is surprising given that the P cells are multipotent. In addition, we know of no verified reports that pluripotent (rather than multipotent) cells are retained in the normal adult as the second sentence suggests.

---

## [Author Response]

Essential revisions:

*Overall this is a compelling piece of work, but there are many places where the manuscript would benefit from improved clarity. […] In addition, since many of the effects shown are partial, quantifications would help to clearly define the impact of the different pathways on D-fate precursors.*

Thank you very much for your involvement in improving the initialversion of our manuscript. We understand that many of the problems with figures/text would have arisen from insufficient data organization, explanations, lower magnification and lack of quantitative data. We have corrected these by reorganizing the figures and supplements, by adding some more schematics, some enlarged pictures and quantifications according to the detailed guidance.

*Finally, virtually all of the genetic analysis is done with mutants that affect many non-airway cells as well as the airway cells, and so many of the effects shown may be indirect, non-cell autonomous effects. For instance the top and btl mutants enhance cell death in a broad swathe of ectodermal cells, and the hh mutants simply lack parts of segments that include airway precursor cells, probably because these cells were simply not generated much earlier in development. These issues might be addressed spatially and temporally targeting the suppression of gene activity to better resolve when and how the noted defects arise in the mutants. The targeted suppression of gene function by RNAi would be of value, particularly of genotypes where the effects of the mutants are likely to be very pleiotropic and there is a possibility that the effects on the trachea are secondary. This is particularly relevant to Figure 6.*

Whether signaling molecules have a direct or an indirect effect on different P/D cell fates in the airway tree is an important issue. Ideally, this could be done with Gal4 drivers expressed very early in airway primordial specification and before/during the P/D fate specification. The widely used btl-gal4 is not efficient for this purpose, which probably reflects that btl-gal4 mediated overexpression of proteins occurs after the invagination of the primordial cells. We tried trh(66)-gal4, GMR15F01(trh), GMR19A07 (vvl), VT041289 (ems) and VT49279 (stg). Except for GMR19A07 (vvl), which is expressed in a different pattern from the reported one, all drove UAS-nGFP mediated GFP expression earlier but much weaker than btl-gal4. Expression of dominant negative form of the effector TF of the hh signaling pathway, cubitus interruptus, ci(rep) with any of these drivers did not give significant effects on the P/D fate selection. We did the same experiments in a sensitized background (vvl/vvl hh mutants) by recombining each of the gal4 lines in hh/vvl mutants. But so far we have not obtained positive results. We also utilized ptc wg double mutant condition where only small number of the P fate cells are specified in the dorsal and ventral stripes of the abdominal segments. Strong overexpression of ci(rep) in the dorsal ectoderm with salm-gal4 locally increased the P-fate cell number in the dorsal stripe implying that hh signaling acts cell autonomously in P/D fate selection (Figure 7). We expect that more rigorous experiments would be possible when a stronger early gal4 driver becomes available.

With our current data, we agree with the reviewers that hh signaling could have non-autonomous effects on the airway development. arm-gal4 mediated overexpression of s-spi in hh mutants results in a continuous band of mab2A12 positive airway cells over many segments (Figure 7—figure supplement 2), which is also seen in wg mutants. dEGFR signaling enhances Wg degradation (Cell, 2001 (105,613-624)). We speculate that overactivation of RTK signaling degraded the potentially remaining small amount of Wg in hh mutants, which results in expansion of the airway primordia.

Hh is known to be a positive regulator of trh expression (Development 128:1599-1606). We agree with the reviewer that there would be some contribution of this hh function to the loss or reduction of the D-fate in hh mutants. However, we think that a more direct role of hh in the D-fate specification needs to be assumed to account for the expansion of the P-fate in hh vvl double mutants and in some metameres of single hh mutants.

In summary, we included the new data in Figure 7 and interpreted that they imply an autonomous role hh signaling in the trachea and also noted that the detailed mechanism of hh and wg signaling would have to await the development of better genetic tools and further experimentation (end of subsection “Wg/WNT and Dpp/BMP signaling converge on the P/D-fate selection”).

*Thus, while the study should prove to be useful, we think it requires some refinement before it is ready for publication.*

*Specific major comments: 1) Figure 1 shows the evolution of marker gene expression in the developing trachea. It would benefit from a clearer diagram indicating the markers that will be used for P and D fate in the subsequent figures.*

We have added a guidance sketch illustrating airway development, the expression of the signaling molecules and the positions of cells acquiring the P/D fates.

*2) The text is somewhat unclear on the relationship between vvl and RTK signalling and at times emphasises the co-operative effects, and at times the data that the RTK ligands are partly downstream of vvl (and Hh). Tightening the summary sentences would help.*

We have tried to be clearer in the text on the relationships of vvl and RTKs (Results and Discussion).

*3) Figure 4. By contrast, the authors show that P fate requires the transcription factor Grn. And that in the absence of the D fate, the Grn domain is expanded and cells acquire the default P fate. Is the D fate at all expanded in the grn mutants? It looks like it is (Figure 4 versus E), but this is not explicit in the text.*

Two scenarios could explain loss of the P fate in grn mutants. The P fate is transformed to the D fate or the P cells are simply lost. Halving the airway cell number in CycA mutant reduced but did not abolish the P0144-lacZ positive cells (Figure 4—figure supplement 1). In contrast, we detected that the Trh positive cell number in the TC/SB region is reduced by around 10 cells in grn mutants (Figure 4). These results suggest that the P fate cells are selectively lost in grn mutants. However, this cannot be explained simply by loss of Trh expression in the P-region because arm-gal4 mediated overexpression of Trh does not restore P0144-lacZ expression in grn mutants (Figure 4—figure supplement 1). Together with the data that grn overexpression expands the P-fate to the D-region (Figure 4), we suggest that grn functions in 2 ways, first to establish Trh expression in the P-region and second to induce the P-fate (subsection “Grn promotes the P-fate selection”).

*4) Figure 5. The epistasis experiments suggest that any residual vvl expression in the grn mutants (e.g. in the grn, Hh double mutants compared to the grn, vvl mutants) is sufficient to prevent P fate development. This strongly suggests that an important role of vvl is to inhibit P fate allowing the D fate to be specified. Indeed, that P is the default fate which must be suppressed by the D-promoting molecules is basically the authors' conclusion. However, the explanation for the effects (that there is substantial vvl expression in the hh mutants) is buried in Figure 3—figure supplement 1. Can the explanation in the text of Figure 5 be improved to make the manuscript easier to read? Given the epistasis experiments in Figure 5, and that P cells can be specified in the absence of grn (e.g. in the grn, vvl double mutants), I don't think it is appropriate to refer to Grn as a master-regulator of P fate in the previous section (Figure 4 description).*

Without D factors, the airway fields uniformly take the P-fate. In this situation, grn becomes indispensable for all trh expression in the main airway, probably through its effect on the maintenance of trh expression (Figure 9). When the D-fate is induced in the anterior-central region of the airway primordia as in wild type, this grn-to-trh regulation is left active only in the P-region. In grn mutants, both the P-fate and the P-region are lost. So we would still prefer to call grn a master regulator of the P-fate.

In grn vvl double mutants, the normal P-region is expected to be lost as in grn single mutants and some of the D-region that is transformed to take the P-fate also would lose trh expression. Accordingly, the trh expressing area is very small in grn vvl double mutants (Figure 5). The P-fate in grn vvl double mutants would have arisen from the normal D-region. We assume that there is another P-fate promoter in the D-region or that the default fate of the D-region is the P-fate. In grn hh double mutants, metameres corresponding to those that lose vvl expression in hh single mutants become grn hh vvl triple mutant, which would lead to loss of trh expression as seen in Tr3 of Figure 5.

*5) Figure 6. Analysis of Wg and Dpp mutants shows that P/D fate is strongly changed in these genotypes. This figure needs additional experiments to clarify the effects of Wg and Dpp. Is A-P and D-V patterning just compromised overall in these mutants, hence very strong effects on P and D fate in the trachea? Or, is there a direct Wg/Dpp signalling input into P and D fate? If RTK or Hh ligand expression (pathways the authors have shown to be required for P/D tracheal patterning) was examined at an earlier stage, would these be altered thus explaining the dramatic changes in the Wg/dpp mutants? An age-matched control shown within Figure 6 (even if repetitive with other figures, and yes, we see that we are contradicting ourselves as we previously asked that redundancy be avoided, but it this instance it will be useful) would assist the reader. (See comments on Figure 9 below). One way to address our concerns here would be to use temporal and tissue specific drivers with RNAi, particularly where concerns on pleiotropic effects can be argued to be of significant concern. Another way, not necessarily excluding this approach would be by live-imaging of phenotypes. In our editorial consultations we concluded that the latter is not a necessary requirement but leave it to the authors to best address this important concern in the manner(s) best suited.*

We have added schematic panels to depict and explain the phenotypes of each mutant (Figure 6). Our aims were initial description of mutant phenotypes regarding the P/D fate discrimination and presentation of possible mechanistic explanation of the phenotypes. We agree that whether the effects of wg/dpp/hh are direct or indirect is an important issue. We added some experiments, our interpretation and stated in the text that this issue is an open question for future experiments until a strong early gal4 driver becomes available.

6) Figure 7 supports the hypothesis from Figure 5 that D-fate determinants suppress the default P fate. Would Figure 7 be better following on from Figure 5?

*Figure 7. It appears as if there is some residual P fate in this picture? Maybe these are cells of another lineage and it would benefit from some arrows, or maybe the text needs modifying?*

We had thought of both figure orders in the initial submission and we had chosen the current figure order. The main reasons for this choice were that wg mutant phenotypes need to be described before hh overexpression experiments due to the hh-wg mutual regulation and that wg mutant phenotypes are better explained together with dpp mutant phenotypes. So instead of changing the figure order, we have tried to make the flow smoother by adding some phrases connecting the text part corresponding to Figure 5 to those corresponding to Figure 7.

Also, we clarified in the figure legend that the aop aos mutant panel (Figure 7 in the revised version and Figure 7 in the old version) shows signals from other lineages.

*7) Figure 8 shows that the various branches of the D part of the trachea respond differently to Dpp signalling, suggesting that the D factors may also affect later tracheal patterning.*

Yes, we agree.

*8) Figure 9. This is a useful diagram but currently a bit complex. Could it be simplified? Perhaps the reader could be taken through the figure more gently by elaborating the legend? (This figure abrogates some earlier comments about Figure 6, though the text still needs to be clarified in that part). Are panels of the figure adapted from Sanson et al. 2001 just depicting their conclusions? In either case the panels should be appropriately referenced.*

We have added a general guidance description and a drawing of embryo segmentation and airway primordia specification in the introduction and Figure 1. We added some schematic panels in Figure 6 to explain the dpp/wg phenotypes. We also tried to make Figure 9 and its legend better. We had referenced Sanson et al. in the initial version of our manuscript as a relatively recent guide for segmentation and parasegment boundaries in the *Drosophila* embryos. Rather than adopting any specific model figures of others, our schematic figures had been produced by us based on our understanding of brilliant work of others on *Drosophila* ectoderm patterning.

*9) Are these phenotypes truly representative of 100% penetrance in every genotype used? Or is there some variation which should be made explicit in the manuscript?*

We have added quantification data to figures and texts.

*10) The Introduction starts with a discussion of vertebrate pluripotency which is surprising given that the P cells are multipotent. In addition, we know of no verified reports that pluripotent (rather than multipotent) cells are retained in the normal adult as the second sentence suggests.*

We have corrected this mistake.